# Urbanization effect on sunshine duration during global dimming and brightening periods in China

Yawen Wang<sup>1,2</sup>, Martin Wild<sup>1</sup>, Arturo Sanchez-Lorenzo<sup>3</sup>, Yonghui Yang<sup>2</sup>, Veronica Manara<sup>4</sup>, and Dandan Ren<sup>2</sup>

<sup>1</sup>Institute for Atmospheric and Climate Science, ETH Zurich, Zurich, 8092, Switzerland
 <sup>2</sup>Center for Agricultural Resources Research, Institute of Genetics and Developmental Biology, Chinese Academy of Sciences, Shijiazhuang, 050021, China
 <sup>3</sup>Instituto Pirenaico de Ecología, Consejo Superior de Investigaciones Científicas (IPE–CSIC), Zaragoza, 50059, Spain
 <sup>4</sup>Department of Physics, Università degli Studi di Milano, Milan, 20133, Italy

Correspondence to: Yawen Wang (yawen.wang@env.ethz.ch)

**Abstract.** There is an ongoing debate on whether the observed decadal variations in surface solar radiation, known as "dimming and brightening", are a global or just local phenomenon. We investigated this issue using a comprehensive set of long-term sunshine duration records from China, which experienced a rapid growth in urbanization during past decades. 172 pairs of urban and nearby rural stations were analyzed over the period 1960–1989 ("dimming phase") and 1990–2013

- ("brightening phase"). There is a large overlap in urban and rural sunshine duration trends for both dimming ( $\approx 86\%$ ) and brightening ( $\approx 84\%$ ) phases. This indicates that rather than urban dimming or rural brightening, the global dimming and brightening phenomena are more of national/regional scale in China. In the dimming phase, sunshine duration significantly declined in both urban and rural areas at an average rate of -0.20 h d<sup>-1</sup> decade<sup>-1</sup> and -0.14 h d<sup>-1</sup> decade<sup>-1</sup> respectively, i.e. rural dimming has been around two-thirds of urban dimming. This ratio generally increases from a minimum of 0.39 to a
- maximum of 0.87 with increasing indices of urbanization, reaching saturation when the urbanization level exceeds 50%, or the urban population exceeds 20 million persons, or the population density becomes higher than 250 person km<sup>-2</sup>. Urbanization can be treated as a useful indicator for anthropogenic air pollution in studying pollution-driven changes in sunshine duration during the dimming phase when pollution control and monitoring were largely absent. After the transition into the brightening phase, the increasing number of environment-related laws and regulations as well as investments in the
- abatement of environmental pollution might have helped in counteracting air pollutants generated during the urbanization process. Therefore, in the brightening phase, urbanization no longer simply indicates an increase in air pollution and its effect on sunshine duration becomes insignificant. In conclusion, urbanization can give a general indication of pollutiondriven sunshine dimming until pollution regulations become effective.

#### **1** Introduction

In line with intensifying anthropogenic disturbances, a decreasing trend in surface solar radiation was noted at widespread locations on the globe between the 1950s and the 1980s with a range of -3 to -9 W m<sup>-2</sup> (Gilgen et al., 1998; Stanhill and

5

Cohen, 2001; Liepert, 2002; Ohmura, 2009; Wild, 2009, 2012). This phenomenon is popularly known as "global dimming", where "global" is originally referred to "global radiation", a synonym for surface solar radiation, yet is often interpreted in terms of a global-scale dimension (Wild, 2009, 2012). Alpert et al. (2005) argued that the observed solar dimming is dominated by the large urban sites, and thus is more of a local rather than a global phenomenon. They further estimated that the decreasing rate in solar radiation fluxes accelerates from -0.05 to -0.32 W m<sup>-2</sup> yr<sup>-1</sup> with population density increasing

- from 10 to 200 person km<sup>-2</sup> (Alpert and Kishcha, 2008). Similarly, Robaa (2009) observed in the Greater Cairo region that the radiation loss in the atmosphere over the urban area was always higher than over the rural area, but the dimming rate during the 1969–2006 period is similar for the urban and rural areas at -0.09 and -0.10 MJ m<sup>-2</sup> yr<sup>-1</sup>, respectively. Based on measurements in Israel between 1954 and 2007, Stanhill and Cohen (2009) argued that the population density was not a
- 10 robust proxy for global radiation change. Liley (2009) found an inaccuracy in the equation used by Alpert et al. (2005) for calculating cumulative mean slope, and thus questioned the conclusions made in Alpert and Kishcha (2008). To ensure that only urban and rural sites under similar climate conditions are compared, Wang et al. (2014a) restricted the analysis to urban-rural station pairs within  $2^{\circ} \times 2^{\circ}$  areas and noted that the average urbanization effect on the mean and the trend of surface solar radiation over Europe, China, and Japan from 1961 to 1990 was small. Questioning the existence of an
- urbanization effect on surface solar radiation, Imamovic et al. (2015) further examined the population indices (PI) at 157 15 worldwide sites for the 1960s–1980s and concluded that the urbanization effect based on PI is insignificant in Europe and Japan but cannot be ruled out in China and Russia. A similar conclusion was made in China by Wang et al. (2012b) using diurnal temperature range (DTR) as a proxy for solar radiation, namely that urbanization effects exist not only during the global dimming period but also in the brightening period. The term "brightening" has been coined to describe the transition
- from a decrease to an increase in surface solar radiation that occurred in many regions of the world from the mid-1980s on 20 the order of 1 to 4 W  $m^{-2}$  (Wild et al., 2005; Wild, 2012, 2016).

The potential correlation between urbanization and solar radiation variations does not only have a statistical meaning but also a physical background. Urbanization, which is accompanied by the industrialization process, plays an important role in producing anthropogenic aerosol/pollutants in the atmosphere. Aerosols can attenuate solar radiation reaching the Earth's surface by scattering/absorbing radiation due to the nature of their composition (direct radiative forcing), or by increasing

25 cloud reflectivity and lifetime through acting as cloud condensation nuclei and ice nuclei (indirect radiative forcing) (Charlson et al., 1992; Ramanathan et al., 2001; Lohmann and Feichter, 2005; Wild, 2012). Direct measurements of aerosol optical depth (AOD) and air pollution index (API) mostly only started during the brightening phase, which limits an adequate investigation of the role of aerosols in the global dimming. Therefore, there is a necessity for an indicator of air 30 pollution conditions also during the global dimming period.

Population density (PD) is the most widely used index to quantify the impact of urbanization on solar radiation in recent studies (Alpert et al., 2005; Alpert and Kishcha, 2008; Stanhill and Cohen, 2009; Imamovic et al., 2015). Besides PD, this research introduces three other indices to represent the magnitude and process of urbanization: urbanization level (ULE), urbanization speed (USP) and urban population (UP). Furthermore, sunshine duration is used as the proxy for surface solar

radiation. Compared with solar radiation, sunshine duration has a much wider spatial and temporal coverage and is almost free of inhomogeneity in China (Xia, 2010; Wang and Yang, 2014; Wang et al., 2015). It has been demonstrated that sunshine duration is able to capture variations in cloudiness and signals from aerosol concentration (Wang et al., 2012c; Sanchez-Romero et al., 2014; Wild, 2016). Nevertheless, the existence of an urbanization impact on the trend of sunshine duration remains unclear.

The main objective of this study is thus to answer the following 3 questions: (1) Is sunshine dimming and brightening in China a local or large-scale phenomenon? (2) Does an urbanization effect on sunshine duration exist in both global dimming

and brightening phases? (3) How does urbanization affect rural and urban sunshine dimming?

#### 2 Data and Methods

- The datasets for yearly sunshine duration and daily total cloud cover for 1961–2013 were collected from the China Meteorological Data Sharing Service System (http://data.cma.cn), governed by the China Meteorological Administration (CMA). Basic quality control on the data (checks on the climatic range and extremum, logical rules, internal and time consistencies) were performed by the CMA (2007). The instruments for measuring sunshine duration in China are Jordan and Campbell-Stokes sunshine recorders. During the period studied here, there was almost no instrument replacement, it
- occurred only in < 2% of the metrological stations (Tao et al., 1997; Wang et al., 2015). Total cloud cover was visually estimated by experienced observers based on the standards of the World Meteorological Organization (WMO). The dimming and brightening phases were defined as the periods of 1961–1989 and 1990–2013 respectively, according to the change-point in the early 1990s in sunshine duration trends in China evidenced by recent studies (e.g., Xia, 2010; Tang et al., 2011; Wang et al., 2013; Wang and Yang, 2014).</p>
- Urban/rural stations were selected according to the administrative divisions of China. There are five administrative levels of local government in China: the provincial, prefecture, county, township and village (http://www.gov.cn/). In this study, meteorological stations located in the provincial/prefectural-level divisions were defined as urban stations, while those located in county/township/village-level divisions were deemed as rural stations. Special cases are county-level cities and city-governed districts, which belong to county-level divisions but were regarded as urban stations in this study for two
- reasons. First, a rapidly accelerating process of urbanization can be expected in county-level cities and city-governed districts, which are mainly created by replacing counties. Besides, county-level cities and city-governed districts are mainly located in/near to prefectural-level cities, and therefore might be frequently influenced by urban pollution or become the source of pollution emissions as a result of industry transfer.
- The raw dataset encompasses 906 stations across latitudes 16°32′–52°58′ N, longitudes 75°14′–132°58′ E, and 30 elevations 2–4800 m. Only stations with sunshine duration records covering at least 85% (≥ 45 years) of the period 1961– 2013 were considered. Then, urban-rural station pairs were strictly picked out for each of the studied 28 provincial-level divisions across China, depending on (1) the administrative divisions (i.e., the selected county is under the jurisdiction of the

corresponding city) and/or (2) location (i.e., the urban-rural station pairs are within  $1^{\circ} \times 1^{\circ}$  and of similar elevation). The studied 28 provincial-level divisions include 22 provinces across the Mainland China, 5 autonomous regions and 1 municipality (Chongqing) (Fig. 1). The other three municipalities of Beijing, Shanghai and Tianjin, where only urban stations meet the criteria of having  $\geq$  45 years' data, were excluded from this study due to a lack of corresponding rural stations

5 stations.

20

25

In the end, 172 urban-rural station pairs all over China were chosen with an average difference in latitude, longitude and elevation of 0°N, 0°E and -95 m, respectively (Fig. 1). 19 of the selected stations (~5.5% of total) were replaced by/replaced another station with similar location, climatic and administrative condition. They were combined to complete the data period. After taking this into account, 98% and 94% of the selected stations have  $\geq$  50 years' data for sunshine duration and total

- 10 cloud cover, respectively. The records of stations with less than 3 years of missing data were filled based on linear regression relationships established from data from other years of the same station separately for the dimming/brightening phase. This aimed to enable comparison in each pair of urban-rural stations for each year with no change in the original trend slope. While the records of stations with more than 3 years of missing data were filled using the data from the nearest station in the same administrative level, which was multiplied by a ratio of the mean value in the test station to that in reference station
- 15 calculated over the period with available data in both stations. In this way, the effect of the different geographic location between the test and reference stations on the gap filled data could be reduced.

Urban and total population of the mainland of China and each provincial division for 1961–2013 was collected from the China Statistical Yearbook, the China Compendium of Statistics, and the Statistical Yearbooks of each province. Then, for each province, urbanization level, population density, and urbanization speed were calculated to quantitatively define urbanization. Urbanization level (*ULE*, %) is determined as follows:

$$ULE = \frac{UP}{TP} \times 100\%, \qquad (1)$$

where UP and TP are urban and total populations, respectively. Population density (PD, person km<sup>-2</sup>) is calculated as:

$$PD = \frac{TP}{LA} \times 100\% , \qquad (2)$$

where *TP* and *LA* are total population and land area ( $km^{-2}$ ), respectively. The speed of urbanization (*USP*, % year<sup>-1</sup>) is computed as follows:

$$USP = \frac{1}{n} \times \left( ULE_{t+n} - ULE_t \right), \tag{3}$$

where *n* is the number of years;  $ULE_t$  and  $ULE_{t+n}$  are the urbanization levels for the years *t* and t+n, respectively. During the 1980s, there is a change in the standard for calculating population in China from registered to permanent residents. This might affect the accuracy of the calculated *USP* for the 1961–2013 period, but still works for comparisons among the

30 regions/provinces. Besides, there is one exception in the provincial *USP* calculation, namely that the Chongqing and Sichuan provinces were combined together. This is because Chongqing used to be part of Sichuan, so that the population of Chongqing and Sichuan cannot be separated before 1997.

## **3 Results and discussions**

## 3.1 Is sunshine dimming and brightening in China a local phenomenon?

To address the question of whether China's dimming and brightening is a local or national/larger scale phenomenon, sunshine duration trends in 172 rural-urban station pairs across China were examined during the global dimming and brightening periods (Fig. 2). From Fig.2A it can be seen that sunshine duration declines at about 90% of the urban stations in the dimming phase, suggesting a nationwide urban dimming. Nevertheless, sunshine dimming is not limited to urban areas but is also noticeable in rural area. Fig.2B shows that decreasing sunshine duration trends also cover about 80% of the rural stations. Rural and urban dimming occur in most of China and generally in the same regions, which indicates that sunshine

dimming in China is more of a large scale rather than a local phenomenon. This conclusion can be further verified from Fig.

- 3A, which shows that there is a large overlap ( $\approx$  86%) between rural and urban sunshine duration change rates. On average, sunshine duration significantly declined in both urban and rural areas at the rate of -0.20 h d<sup>-1</sup> decade<sup>-1</sup> and -0.14 h d<sup>-1</sup> decade<sup>-1</sup>, respectively (Fig. 4A). Although the dimming rate at the rural sites is lower than at the urban sites, this yet provides no convincing evidence that sunshine dimming is exclusively in urban area and hence a local phenomenon. The averaged urban dimming rate in China is slightly lower than that obtained in previous studies (Wang et al., 2013; Wang and
- Yang, 2014), because a few mega cities without available nearby rural stations were excluded from this study.

Unlike in the dimming phase where the decline in sunshine duration is almost nation-wide distributed, China's brightening depicts a more complicated spatial pattern. From Fig. 2C and 2D it can be inferred that in the brightening phase, an increase in sunshine duration occurs in about half of China, especially in the regions of southern, northwestern and northeastern China, while dimming continues in the other half, especially in the North China Plain, where haze pollution is

20 being reported (Wang et al., 2014c). This spatial pattern of sunshine duration trends is obtained in both rural and urban areas, suggesting a regional rather than local brightening. Rural and urban sunshine duration changing rates overlaps by about 84% in the brightening phase (Fig. 3B). Averaged over the whole China, the sunshine duration trend in both urban and rural stations levels off to -0.01 h d<sup>-1</sup> decade<sup>-1</sup> for 1990–2013 (Fig. 4A).

In summary, the global dimming phenomenon in China is not dominated by the large urban sites but also exists in rural areas. In the dimming phase, both urban and rural sunshine duration significantly declined. Similarly, in the brightening phase, sunshine duration recovery happened not only in rural areas, which are supposed to be less developed and polluted, but also in the big cities. Therefore, it can be summarized that sunshine dimming and brightening in China is not a local but more of a large-scale phenomenon.

#### 3.2 Existence of urbanization effect on sunshine duration in global dimming and brightening phases

The urbanization process is highly correlated with the increase in energy consumption, industrial GDP (Gross Domestic Product) and civil vehicles (Table 1), which are the major anthropogenic sources of air pollution emissions. For 1960–2013,

15

urbanization was continuously progressing in China with three stages (Fig. 5), in line with the historical periods of social and economic developments in China (Shiu and Lam, 2004).

Before 1978, sunshine duration declined by  $0.10 \text{ h} \text{ d}^{-1} \text{ decade}^{-1}$  averaged over the 344 stations across China (Fig. 4A). Meanwhile, total cloud cover trends stabilized by 0.3% decade<sup>-1</sup> for 1960–1978 (Fig. 4B), which cannot fully explain the

- obvious decrease in sunshine duration. The urbanization level (*ULE*) was low and stable (Fig. 5), but the total population of China started to increase from 1960 by an almost linear trend of 185 million persons decade<sup>-1</sup>. Increases in pollution emissions can be expected also from the industrial growth during the pre-reform period shown in Fig. 6A, which was in an extensive way characterized by low efficiency of energy use (Zhang, 2005; Fei et al., 2011). Furthermore, environmental protection was of seldom concern in this period, and the first law with legal provisions on environment protection was issued
- only after 1978 in China as indicated in Fig. 6B.

After the adoption of the open door policy in 1978, the first notable increase in *ULE* happened with a rate of 6.5% decade<sup>-1</sup> for 1978–1995 (Fig. 5). During this period, the number of laws and regulations on environment protection gradually increased from 0 to 34 (Fig. 6B), suggesting a beginning, even if not so efficient, of pollution control. Thus, the strongest decline was noted in sunshine duration in the 1980s, thereafter sunshine duration stabilized at the lowest level of the past half century (Fig. 4A).

From 1995 onward with the transition from extensive to intensive growth, the increasing rate of *ULE* in China doubled to 13.7% decade<sup>-1</sup> (Fig. 5). However, the sharper increase in *ULE* does not result in a stronger decrease in sunshine duration but in contrast in a levelling off by -0.01 h d<sup>-1</sup> decade<sup>-1</sup> in both rural and urban areas (Fig. 4A), suggesting an insignificant urbanization effect on sunshine duration in the brightening phase. Effective air pollution regulations after 1995 are indicated

- in Fig. 6B, in that the number of environment-related laws and regulations rapidly increased to 158 in the year 2013. Besides, the investment completed in the treatment of environmental pollution in China in 2013 (952 billion yuan) was 9.4 times of that in 2000 (101 billion yuan) (Fig. 6C). The increasing pollution control may have helped to offset anthropogenic air pollution induced during the urbanization process, so that the API (Air Pollution Index) of China has declined in the 2000s (Lei et al., 2011; Wang et al., 2012a; Wang et al., 2013). Urbanization can give a general indication of pollution background
- not only in urban but also in rural areas until pollution regulations become effective.

Overall, urbanization effects on sunshine duration variations exists in the dimming phase in China, consistent with the findings of Wang et al. (2012b) based on DTR and Imamovic et al. (2015) based on surface solar radiation records. Cloud effects could not meaningfully contribute to the decadal variations in rural and urban sunshine duration in the dimming phase for two reasons: (1) both rural and urban total cloud cover slightly declined in the dimming phase of 1960–1989 (Fig. 4B),

which cannot result in a decrease in sunshine duration; (2) the difference in urban and rural cloudiness remains similar in the dimming phase, which would not cause a widening urban-rural contrast in sunshine duration. Please note that the average difference in latitude, longitude and elevation between the selected urban and rural stations also won't lead to a varying difference in rural and urban sunshine duration. The urbanization effect on sunshine duration is less significant in the brightening phase, when urbanization no longer simply indicates an increase in air pollution as a result of strengthened

pollution regulations. The cloud trend couldn't be an explanation for the leveling off in sunshine duration trend for 1990–2013 either, as both rural and urban total cloud cover trends significantly increased (Fig. 4B). The conclusion of an insignificant effect of urbanization in the brightening phase differs from that made by Wang et al. (2012b) based on DTR in China. They found a decreasing trend in urban DTR and an increasing trend in rural DTR in the brightening phase, where urban and rural areas are represented by 71 large cites, 234 small cities and 97 rural stations defined by urban population (UP)  $\geq 0.5$ , 0.05~0.5 and < 0.05 million persons, respectively. Rural and urban stations defined in the study of Wang et al. (2012b) are not equally distributed across China with most rural stations in the western part and most urban stations in the eastern part, and thus are faced with quite different conditions in terms of climate and development.

## 3.3 Quantifying urbanization effects on rural and urban sunshine dimming

After clarifying the existence of urbanization effects on sunshine duration in the dimming phase, we attempted to further quantify the effect of urbanization on sunshine dimming. For this purpose, China was divided into regions with different scales of urbanization level (*ULE*), urbanization speed (*USP*), urban population (*UP*) and population density (*PD*) (Fig. 7). The cut-off points were chosen based on the quantile method, in order to ensure equal number of provinces in each divided scale. *ULE*, *UP* and *PD* for each province were calculated based on the data for 2013, which was assumed to be able to reflect the general condition of each province for the past decades for comparisons.

Comparing Fig. 2A and 2B with Fig. 7, exceptions of increasing sunshine duration trends in the dimming phase mainly distribute in less-urbanized provinces, while strong declines in sunshine duration generally occur in the urbanized provinces. This is further illustrated in Fig. 8, which shows that the dimming rate in both urban and rural sunshine duration initially strengthens with increasing *ULE*, *UP* and *PD*, but saturates when ULE > 50%, or UP > 20 million persons, or PD > 250

- person km<sup>-2</sup>. Sunshine dimming shows a different relation with *USP*, which is the only index taking the urbanization condition in the 1960s into account. Seen from Fig 8B, rural and urban dimming strengthens when *USP* increases from < 0.6% year<sup>-1</sup> to 0.6~0.7% year<sup>-1</sup>, but obviously weakens when *USP* further increases into 0.7~0.78 % year<sup>-1</sup>, and then restrengthens when *USP* is greater than 0.78 % year<sup>-1</sup>. The reason for the main exception in the category of *USP* of 0.7~0.78 % year<sup>-1</sup> is indicated in Fig. 7, which shows that half of the provinces with *USP* of 0.7~0.78 % year<sup>-1</sup> have an urbanization
- level only of 45~50% in 2013. Therefore, even with a relatively higher *USP*, these provinces are still less-urbanized and thus a relatively weaker dimming trend can be noted there (Fig. 2).

On average, the decline in sunshine duration in rural area ( $-0.14 \text{ h} \text{ d}^{-1} \text{ decade}^{-1}$ ) is about two-thirds of that in urban area ( $-0.20 \text{ h} \text{ d}^{-1} \text{ decade}^{-1}$ ) in the dimming phase (From Fig. 4A). This ratio changes with different scales of *ULE*, *UP* and *PD* in a quadratic function with a parabola opening downward (Fig. 8). The ratio of rural to urban dimming gradually intensifies

from 0.54 to 0.85 when *ULE* increases from <45 to 50~55%, from 0.47 to 0.87 when *UP* increases from <14 to 20~35 million persons, and from 0.39 to 0.86 when *PD* increases from <100 to 250~370 person km<sup>-2</sup>, and afterwards slightly decreases with further increases in these urbanization indices. By contrast, the rural-urban dimming ratio generally shows a

linear relation with *USP*, changing from 0.52 to 0.83 with *USP* accelerating from <0.60 to >0.78 % year<sup>-1</sup>, due to the decrease of the ratio in the *USP* category of 0.7~0.78 % year<sup>-1</sup>.

With increasing urbanization, rural dimming generally becomes more close to urban dimming, possibly due to industrial transfer (Li, 2010) as well as wind dispersion of pollutants/aerosols from urban to nearby rural areas (Wang et al., 2014b; Yang et al., 2009). This effect saturates when urbanization reaches certain level (ULE > 50%/UP > 20 million persons/PD > 250 person km<sup>-2</sup>), above which a further increase in urbanization won't generate a further alignment of urban and rural dimming, as well as a further strengthening of these trends. Similarly, Imamovic et al. (2015) questioned the existence of an urbanization effect on surface solar radiation in the well urbanized regions of Europe and Japan but suggested that such an effect cannot be ruled out in Russia and China. It is assumable that the government interference with regulations on environmental pollution might start first in relatively urbanized/developed regions.

## **4** Conclusions

Based on sunshine duration data from 344 sites across China, the effect of urbanization on sunshine duration was evaluated in the global dimming (1960–1989) and brightening (1990–2013) phases, respectively. In general, the urbanization impact has two aspects: (1) it describes the general increase in air pollution emissions which leads to a decrease in sunshine duration in both urban and rural areas, and (2) may lead to a difference between rural and urban sunshine duration trends.

The global dimming phenomenon was proven to be of large (national) scale in China based on three lines of evidence: (1) the declining trend in sunshine duration in China was significant in both rural and urban areas in the dimming phase; (2) spatially, negative sunshine duration trends cover about 90% of the 172 urban stations and 80% of the 172 nearby rural stations; (3) a large overlap ( $\approx 86\%$ ) exists in the rates of sunshine duration changes between the selected rural and urban

- stations across China. Sunshine brightening, which covers about half of China and which is almost equally distributed in both rural and urban areas, can also be considered as a regional phenomenon. The changing rates of rural and urban sunshine duration overlap by about 84% in the brightening phase. Therefore, rather than urban dimming or rural brightening, the global dimming and brightening phenomena in China is evident on a large spatial scale.
- Sunshine duration variations are sensitive to changes in urbanization in the dimming phase. Urbanization as a proxy for anthropogenic air pollution is applicable in the periods without efficient air pollution control, which corresponds also to the period when air quality monitoring was insufficient. During the brightening phase, air pollutants generated in the urbanization process might have been partially counteracted by the increasing regulations on environment and investments in the abatement of environmental pollution. As a consequence, urbanization effect on sunshine duration trend becomes insignificant. The value of urbanization in analyzing air pollution-induced changes in sunshine duration is emphasized in the 30 global dimming phase until its transition into brightening.

Averaged for the whole China, rural dimming is two-thirds of urban dimming, indicating an overestimation of China's dimming when only urban-scale sunshine duration sites are considered. The ratio overall changes from the minimum of 0.39

5

to the maximum of 0.87 under different conditions of urbanization. It generally increases in more urbanized regions where rural areas might be more influenced by urban pollution, and saturates when urbanization reaches certain level (urbanization level > 50%/urban population > 20 million persons/population density > 250 person km<sup>-2</sup>). The ratio of rural to urban dimming shows a quadratic relation (parabola opening downward) with urbanization, quantified by the indices of urbanization level, urbanization population and population density. Urbanization can be treated as an indicator for

anthropogenic pollution when studying aerosol effect on sunshine dimming.

## Acknowledgments

This research was accomplished under the support of the Swiss Government Excellence Scholarship (Reference No. 2015.0409/China/OP) cooperated between the FCS (Federal Commission for Scholarships for Foreign Students) and the

10 2015.0409/China/OP) cooperated between the FCS (Federal Commission for Scholarships for Foreign Students) and the CSC (China Scholarship Council). The paper is funded by the National Natural Science Foundation of China (Grant No. 41501036). The co-author Arturo Sanchez-Lorenzo is supported by a postdoctoral fellowship JCI-2012-12508, and projects CGL2014-55976-R and CGL2014-52135-C3-1-R financed by the Spanish Ministry of Economy and Competitiveness.

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
