# Peer review of "Urbanization effect on sunshine duration during global dimming and brightening periods in China"

_Atmospheric Chemistry and Physics, 2016_

## Referee Comment (RC1) · KT Tanaka (Referee) · 8 Sep 2016

Overall comments

This paper investigates if and how air pollution might have affected the sunshine duration records in China over the past several decades. This study particularly looked into records at pairs of urban and rural stations that are geographically proximate each other. The authors identified factors that explain the decadal trends of sunshine duration records at different locations in China by using several proxies for urbanization including population density, a few related constructs, industrial GDP, and related investments.

[Figure]

I enjoyed reading this paper, and I think this will potentially be a good piece of work. As far as I follow the literature, most of the conclusions derived from this study have also been shown by several previous studies (e.g. (Wang et al. 2012a; Wang et al. 2012b; Wang et al. 2013; Wang et al. 2014)). This study strengthens such earlier conclusions based on a newly performed analysis from various angles, providing deeper insights into what might have caused the changes in sunshine duration and also surface solar radiation records in China over the past several decades.

As a major comment, I only take issue with their argument related to the possible effect of clean air policies on sunshine duration records. The authors claim that changes in sunshine duration records since 1995 can be partly explained by recent gradual penetration of pollution regulations. Their claim is based on the number of laws and regulations for "environmental protection" in China since 1978 (Fig 6B). The paper argues that the recent emergence of environmental policies in China may explain the plateau in the trend of sunshine duration records over the last two decades there.

For this argument, I raise two issues as follows: First, no specific information is given for such laws and regulations, except for the link to the website of the Ministry of Environmental Protection of the People's Republic of China. I wonder if they are all really related to air pollution. Could it be that they include those related to eutrophication, for example? A clarification is needed here. Second, in contrast to what the authors derive from Fig 6B, severe air pollution in major Chinese cities is a globally well-known "current" issue that has been worsened or at least persisting over the past few decades. Data for most pollutant and aerosol precursor emissions show no clear indication for their declines (e.g. (Ohara et al. 2007; Lu et al. 2011; Lin et al. 2014)). Given this, I am not entirely convinced by the claim that pollution control in China contributed to the recent flat trend of sunshine duration there. Then, what would explain such a plateau in the sunshine duration trend? To me, this is an open question. Factors are needed to offset the rising trend of cloud cover over the past two decades. But I would think it is not clean air policies since the emission data indicate otherwise.

Overall, I suggest a revision to be further considered for publication in Atmospheric Chemistry and Physics. I have more specific comments as detailed below. With respect to writing, I do follow most part of the paper, but I think some text editing including English editing would help enhance the clarity of the paper.

Specific comments

Page 1, Lines 15-16 From the abstract alone, it is not clear what the numbers (86% and 84%) indicate. It is also unclear what "a large overlap" means.

Page 1, Lines 27-28 Because of the reasoning in the overall comment above, I do not think that this claim is substantiated.

Pages 2 and 3 In spite of the conflict of interest, I cannot ignore my recent work (Tanaka et al. 2016), which deals with the very questions addressed in this study. Please consider incorporating (Tanaka et al. 2016) in the discussion if you agree to do so.

Page 2, Line 11 As far as I can see, (Liley 2009) does not assert the fact that the slope-related problem found in (Alpert et al. 2005) persists in (Alpert and Kishcha 2008). (Liley 2009) raises a different problem for (Alpert and Kishcha 2008).

Page 3, Lines 1-4 The discussion here is a little too short, I think. It would be helpful if the authors discuss some more issues when one compares sunshine duration records with surface solar radiation measurements. As the authors wrote, sunshine duration records have a wider spatial and temporal resolution, which is a clear advantage, given the lack of surface radiation data. But are they almost always consistent with surface solar radiation records?

Page 3, Lines 6-8 From reading it, it does not come clear to me why these questions are worthy of exploration. I suggest that the authors elaborate a bit more to convince readers of the importance of such questions. Also missing are some clear statements on what are actually new in this paper. There are closely related studies like (e.g. (Wang et al. 2012a; Wang et al. 2012b; Wang et al. 2013; Wang et al. 2014)) as cited

in this paper. Some specific statements on what are different from previous studies would clarify the value of this study. This is a question of writing style, but it is usually more common to write "three questions" than "3 questions."

Page 3, Lines 14-15 This sentence structure needs to be fixed.

Page 4, Line 7 A few more digits should be shown for 0°N and 0°E? Otherwise, these are not very useful.

Page 4, Lines 7-8 Why are the 19 stations replaced?

Page 4, Line 23 "100%" on the right hand of equation (2) probably mistakenly entered into the equation.

Page 4, Line 24 Please fix the unit for land area.

Page 5, Section 3.1 Previous studies need to be integrated in the discussion here because this conclusion has been shown by several others.

Page 5, Line 30 Does it sound better if "highly correlated" is replaced by "accompanied by"?

Page 7, Line 19 The data suggest "decrease" rather than "saturate."

References

Alpert P, Kishcha P (2008) Quantification of the effect of urbanization on solar dimming. Geophys Res Lett 35 (8):L08801. doi:10.1029/2007gl033012

Alpert P, Kishcha P, Kaufman YJ, Schwarzbard R (2005) Global dimming or local dimming?: Effect of urbanization on sunlight availability. Geophys Res Lett 32:L17802. doi:10.1029/2005gl023320

Liley JB (2009) New Zealand dimming and brightening. Journal of Geophysical Research: Atmospheres 114 (D10):D00D10. doi:10.1029/2008JD011401

Lin J, Pan D, Davis SJ, Zhang Q, He K, Wang C, Streets DG, Wuebbles DJ, Guan D

(2014) China's international trade and air pollution in the United States. Proceedings of the National Academy of Sciences 111 (5):1736-1741. doi:10.1073/pnas.1312860111

Lu Z, Zhang Q, Streets DG (2011) Sulfur dioxide and primary carbonaceous aerosol emissions in China and India, 1996–2010. Atmos Chem Phys 11 (18):9839-9864. doi:10.5194/acp-11-9839-2011

Ohara T, Akimoto H, Kurokawa J, Horii N, Yamaji K, Yan X, Hayasaka T (2007) An Asian emission inventory of anthropogenic emission sources for the period 1980–2020. Atmos Chem Phys 7 (16):4419-4444. doi:10.5194/acp-7-4419-2007

Tanaka K, Ohmura A, Folini D, Wild M, Ohkawara N (2016) Is global dimming and brightening in Japan limited to urban areas? Atmos Chem Phys Discuss 2016:1-50. doi:10.5194/acp-2016-559

Wang K, Ma Q, Wang X, Wild M (2014) Urban impacts on mean and trend of surface incident solar radiation. Geophys Res Lett 41 (13):4664-4668. doi:10.1002/2014GL060201

Wang K, Ye H, Chen F, Xiong Y, Wang C (2012a) Urbanization Effect on the Diurnal Temperature Range: Different Roles under Solar Dimming and Brightening. J Clim 25 (3):1022-1027. doi:doi:10.1175/JCLI-D-10-05030.1

Wang Y, Yang Y, Han S, Wang Q, Zhang J (2013) Sunshine dimming and brightening in Chinese cities (1955-2011) was driven by air pollution rather than clouds. Clim Res 56 (1):11-20

Wang Y, Yang Y, Zhao N, Liu C, Wang Q (2012b) The magnitude of the effect of air pollution on sunshine hours in China. Journal of Geophysical Research: Atmospheres 117 (D21):n/a-n/a. doi:10.1029/2011JD016753

---

## Referee Comment (RC2) · Anonymous Referee #2 · 26 Sep 2016

Review of Manuscript entitled "Urbanization effect on sunshine duration during global dimming and brightening periods in China" by Wang et al. Dimming and brightening are interesting phenomena observed across the world, which were widely reported in the publications. Causes for these phenomena and implications for climate change were also widely discussed. China has experienced some interesting features in the long-term variation of surface solar radiation and much attention has been paid to this issue. Using polulation as an indicator, the authors discussed the urban and rural differences in the trend of sunshine duration across China. The effect of urbanization on the dimming and brightening was discussed in detail. Some interesting results were presented in this manuscript. This manuscript may be improved and my major concerns are as

follows. 1. The major finding of this study is that urbanization may play an important role in the dimming period, but not in the brigntening period in China. But as suggested by the authors, the urbanization level was low and stable (P6, L5) before 1978, which was accompanied by a significant decreasing trend of sunshine duration. This looks some controversy to the conclusion. 2. The authors argued that urbanization might not be able to reflect variation of atmospheric enironment sine 1990s because regulations were gradually taken into action. This is somewhat speculative and needs some evidence. Satellite aerosol data such as MODIS AOD products during last 16 years may be used to shed light on this issue, otherwise, the speculation is not acceptable. 3. The authors used sunshine duration as a proxy data for surface solar radiation. It should be noted that sunshine duration mainly reflect cloud information. It may be argued that long-term variation of cloud cover did not support that of sunshine duration. However, cloud cover can not reflect all effects of cloud on surface solar radiation, furthermore, long-term trend of cloud cover obtained from surface manual observations is not free of large uncertainty. Therefore, it is suggest to exclude cloud effect on surface solar radiation in the analysis, for example, to perform similar analysis based on sunshine duration measurements in the case of cloud cover of 20%. 4. There are some stations showing long-term trend of sunshine duration quite different from that obtained in the majority of stations (Figure 2), some discussion on this issue is required. This might be related to cloud variation since the general tendency in the urbanization in China should be not quite different. 5. The manuscript should be polished in language.

Some minor issues. 1. Abstract, this ratio should be defined. 2. Why the urbanization effect on dimming diminished when urbanization level exceeds 50%. I wonder whether there are regions where the urbainization level reached to this level in the eastern China before 1990s. 3. P2, L24, delete due to the nature of their composition 4. P2, L35, delete furthermore 5.P3, L3, since sunshine duration is related to cloudiness, why is is excluded in the analysis 6. P3, L10, urbanization also occurs in the rural regions? 7. P4, L20-25, what's difference between county and county-level cities 8. P4, L3-5, i do not understand why three big cities were excluded in the analysis since they

may be use good examples for showing urbanization effect on surface solar radiation. This, certaintly, is related to the method, I do not understand why the analysis were performed in each province. 9. P4, L7-8, not understood. 10. Annual ULE, PD, USP data? 11. Why the trends derived in rural and urban stations did not overlap in 14% pairs. 12. P5, L 16-20, Since sunshine duration remains to decrease slightly in China, it seems not suitable to say brightening. 13. P6, L3-5, total cloud cover stabilized or decreased? 14. P6, L26-30, it should be cloud cover effect, not cloud effect. I do not agree with this discussion on cloud effect. 15. P7, L5-10, It seems in the Wang et al. study, rural-urban pairs are discussed in the 5*5 degree grid.

---

## Author Comment (AC1) · 21 Oct 2016

Dear Dr. Katsumasa Tanaka,

Thank you so much for taking time and energy in reviewing and improving this paper. We are delighted to receive your insightful suggestions. We also thank you for the positive comments on this study.

Your main concern is about the statement on the effective air pollution regulations in China after 1995. Regarding this, you raised two major issues. The first is about the specific aspects of the laws and regulations mentioned in Figure 6. With your question, we will double check the detailed information on all the environmental laws and regulations. Only those having possible effects on air quality protection will be picked out in the revised version. Figure 6B will then be redrawn and the corresponding statement in both caption and text will be modified.

The second major issue is about the air pollution trend in China. We understand your concern. Actually, the globally well-known hazy pollution problem in China is not a nationwide phenomenon. This is indicated in Fig. 2 in that sunshine dimming mainly continues in the area of the North China Plain, where haze pollution is being reported, while brightening is prevalent in South China after 1990. Besides, the severe hazy pollution problem in the North China Plain only occurs in the winter season. The increasing emission trend mentioned in the references of Ohara et al. (2007), Lu et al. (2011) and Lin et al. (2014) is in accordance with the increasing urbanization level in China (Fig. 5), and thus consistent with our main conclusion. In the dimming phase without effective pollution regulations, the emissions generated during the urbanization process were directly changed into equivalent pollutants. By contrast, in the brightening phase, the increasing emissions were compensated by the clean air policies and investments. Therefore, urbanization no longer simply means an increase in air pollution in the brightening phase, and its effect on sunshine duration trends becomes insignificant. But we agree with you that the reason for the recent leveling off in sunshine duration trend in China is still an open question and needs further discussion. We will add a similar discussion in Section 3.2 of the revised manuscript.

In the following context, please find the point-by-point responses to your specific comments.

Comment 1: Page 1, Lines 15-16 From the abstract alone, it is not clear what the numbers (86% and 84%) indicate. It is also unclear what "a large overlap" means.

Reply 1: Thanks for this comment. In the revised manuscript, Lines 15-16 of the abstract will be modified into "Urban and rural sunshine duration trends show similar spatial patterns for the global dimming and brightening phases, respectively."

Comment 2: Page 1, Lines 27-28 Because of the reasoning in the overall comment above, I do not think that this claim is substantiated.

Reply 2: Please refer to the replies to your major comment#2 shown above.

Comment 3: Pages 2 and 3 In spite of the conflict of interest, I cannot ignore my recent work (Tanaka et al. 2016), which deals with the very questions addressed in this study. Please consider incorporating (Tanaka et al. 2016) in the discussion if you agree to do so.

Reply 3: The reviewer is right. We should cite this recent research on urbanization effect. Thank you so much for sharing it with us! We will add this reference at the end of the 1$^{st}$ paragraph of the Introduction section as "A very recent work of Tanaka et al. (2016) put forward a new approach to infer urbanization by combining population data, historical land use maps, satellite images and site visit experiences, and further proved that the global dimming and brightening phenomenon in Japan was not restricted to urban areas and not primarily driven by local air pollution."

Tanaka, K., Ohmura, A., Folini, D., Wild, M., and Ohkawara, N.: Is global dimming and brightening in Japan limited to urban areas?, Atmos. Chem. Phys. Discuss., 2016, 1-50, 10.5194/acp-2016-559, 2016.

Comment 4: Page 2, Line 11 As far as I can see, (Liley 2009) does not assert the fact that the slope-related problem found in (Alpert et al. 2005) persists in (Alpert and Kishcha 2008). (Liley 2009) raises a different problem for (Alpert and Kishcha 2008).

Reply 4: Sorry for this mistake. We will correct this statement by "Liley (2009) disputed the conclusions made by Alpert and Kishcha (2008) and claimed that a large anthropogenic effect in the vicinity of dense population does not negate the finding of long-term change in sparsely populated regions, where a downward trend of −0.16 W m$^{-2}$ year$^{-1}$ was given in their previous work of Alpert et al. (2005)."

Comment 5: Page 3, Lines 1-4 The discussion here is a little too short, I think. It would be helpful if the authors discuss some more issues when one compares sunshine duration records with surface solar radiation measurements. As the authors wrote, sunshine duration records have a wider spatial and temporal resolution, which is a clear advantage, given the lack of surface radiation data. But are they almost always consistent with surface solar radiation records?

Rely 5: Great suggestion! According to your suggestion, we will reinforce this discussion as follows:

"Furthermore, sunshine duration is used as the proxy for surface solar radiation. Compared with solar radiation, sunshine duration has a much wider spatial and temporal coverage and is almost free of inhomogeneity in China (Xia, 2010; Wang and Yang, 2014; Wang et al., 2015). For the whole China (latitudes 16°32′–52°58′ N and longitudes 75°14′–132°58′ E), there are 130 solar radiation stations, with only 59 stations covering at least 85% of the measurement period from 1958 to the present. Moreover, the majority of solar radiation stations are located in urban areas, which will further limit the available data samples for studying urbanization effects. In contrast, there are 906 sunshine duration stations, thus exceeding the solar radiation stations by a factor of about 7. Besides, 638 of them have sunshine duration records which cover at least 85% of the whole measurement period since the 1950s. It has been demonstrated that sunshine

duration is able to capture variations in cloudiness and signals from aerosol concentrations (Sanchez-Lorenzo et al., 2009; Wang et al., 2012c; Sanchez-Romero et al., 2014, 2016; Li et al., 2016; Wild, 2016). Similar with the solar radiation trend, a transition from decreasing to levelling off was also noted in sunshine duration trend in China around 1990 (Wang et al., 2013, 2014). Nevertheless, the existence of an urbanization impact on the trend of sunshine duration remains unclear."

Comment 6: Page 3, Lines 6-8 From reading it, it does not come clear to me why these questions are worthy of exploration. I suggest that the authors elaborate a bit more to convince readers of the importance of such questions. Also missing are some clear statements on what are actually new in this paper. There are closely related studies like (e.g. (Wang et al. 2012a; Wang et al. 2012b; Wang et al. 2013; Wang et al. 2014)) as cited in this paper. Some specific statements on what are different from previous studies would clarify the value of this study. This is a question of writing style, but it is usually more common to write "three questions" than "3 questions."

Reply 6: We agree with you. Based on your comments, we will modify the last paragraph of the Introduction section as follows:

"This study will then make a first attempt to examine the urbanization effect on sunshine duration variations in China. The wide temporal and spatial coverage of sunshine duration observations in China provides a unique opportunity to fully understand the differences of the dimming and brightening phenomenon between rural and urban areas. The value of urbanization as an indicator of pollution level will be evaluated for different stages of social and economic developments in China. In conclusion, the effect of urbanization on sunshine duration variations will be quantified."

Comment 7: Page 3, Lines 14-15 This sentence structure needs to be fixed.

Reply 7: We will modify this sentence into "During the period studied here, instrument replacements only occurred in < 2% of all the meteorological stations (Tao et al., 1997; Wang et al., 2015)".

Comment 8: Page 4, Line 7 A few more digits should be shown for 0° N and 0° E? Otherwise, these are not very useful.

Reply 8: We will modify this sentence into "Finally, 172 urban-rural station pairs all over China were chosen with an average difference of 95 m in elevation and negligible differences in average latitude (~0.001°N) and longitude (~0.002°E)".

Comment 9: Page 4, Lines 7-8 Why are the 19 stations replaced?

Reply 9: Sorry for the confusing statement. We will correct the sentence as "Records from 19 of the selected stations (~5.5% of total) were completed based upon their collocated stations with similar climatic and administrative conditions."

Comment 10: Page 4, Line 23 "100%" on the right hand of equation (2) probably mistakenly entered into the equation.

Reply 10:  You are right. It will be deleted in the revised version.

Comment 11: Page 4, Line 24 Please fix the unit for land area.

Reply 11: Thank you so much for noticing this mistake. We will correct it by "land area ($km^2$)".

Comment 12: Page 5, Section 3.1 Previous studies need to be integrated in the discussion here because this conclusion has been shown by several others.

Reply 12: Yes, you are right. We will add a new sentence (after line 13, Page 5) with more previous works to discuss the conclusions drawn in Section 3.1 as follows:

"Although the dimming rate at the rural sites is lower than at the urban sites, this yet provides no convincing evidence that sunshine dimming is exclusively in urban area and hence a local phenomenon. Dimming in rural/sparsely populated areas was also noted in Europe, Japan, New Zealand, Egypt and the Greater Tel Aviv region in previous studies (Liley, 2009; Robaa, 2009; Stanhill and Cohen, 2009; Wang et al., 2014a; Tanaka et al., 2016). The averaged urban dimming rate in China is slightly lower than that obtained in previous studies (Wang et al., 2013; Wang and Yang, 2014), because a few mega cities without available nearby rural stations were excluded from this study."

Comment 13: Page 5, Line 30 Does it sound better if "highly correlated" is replaced by "accompanied by"?

Reply 13: We will replace "highly correlated with" by "accompanied by" in the revised manuscript.

Comment 14: Page 7, Line 19 The data suggest "decrease" rather than "saturate."

Reply 14: The reviewer is right. The word "saturate" will be replaced by "decrease" in the revised manuscript.

---

## Author Comment (AC2) · 30 Oct 2016

**Reply to Anonymous Referee #2**

We appreciate your insightful suggestions. With your constructive comments, this manuscript, especially Section 3.2, will be largely improved. And further consideration will be given on cloud effects.

Please find the point-by-point responses to your specific comments in the following context.

**Major comments**

1. The major finding of this study is that urbanization may play an important role in the dimming period, but not in the brightening period in China. But as suggested by the authors, the urbanization level was low and stable (P6, L5) before 1978, which was accompanied by a significant decreasing trend of sunshine duration. This looks some controversy to the conclusion.

Reply 1: Thanks very much for noticing this issue. The low and stable urbanization level before 1978 might be due to the official calculation method of urban and rural population. In the urban population only the formally registered urban residents were considered, while people who worked in urban areas without a registered urban residence were still counted as rural population. At that time and even now, it is not so easy for the migrant workers to get a registered residence from the cities they work in. This population calculation method was changed from registered to permanent residents during the 1980s. A similar discussion will be added to the 2nd paragraph of Section 3.2, according to your comments. Since the calculation of urbanization level contains such issues, the bold line in Fig. 5 will be deleted, and accordingly Fig. 5 will be combined with Fig. 4 as Fig. 4C.

An increasing urbanization process during the 1960–1978 period can also be inferred from the GDP composition trends shown in the figure reported below. This figure will replace Fig. 6A of the old version and become Fig. 5A in the revised manuscript (since the former Fig. 5 will become the part C of Fig. 4). The secondary sector of the economy increased by 7.6% decade$^{-1}$ during the 1960–1978 period, whereas the primary sector decreased by 3.3% decade$^{-1}$. This suggests a transfer from primary industry to secondary industry, which is one of the main characteristics of urbanization.

[Figure]

Based on your comments, we will thoroughly modify the second and third paragraph of Section 3.2 as follows:

"In the dimming phase between 1960 and 1989, China experienced a boom in population and industrialization. China's population rapidly increased by an almost linear trend of 166 million persons decade$^{-1}$ (Fig. 4C). The urban population only accounted for a small portion of the total population, and grew slowly during the dimming phase. One possible explanation to this is the calculation method of urban population in China before the 1980s where only the formally registered urban residents were counted in. In this case, people worked in urban areas but without a registered urban residence were still counted as rural population. This standard for calculating population was changed from registered to permanent residents during the 1980s. An increasing urbanization process during the dimming phase can be inferred also from the GDP composition trends (Fig. 5A). The portion of the secondary industry in China's GDP largely increased by 4.2% decade$^{-1}$ during the period of 1960–1989. Meanwhile, the primary sector of the economy decreased by 3.8% decade$^{-1}$. This suggests a transition from primary industry to secondary industry. The secondary industry, which contributed up to 48% to the total GDP, became the backbone of China's booming economy at this stage. The industrial growth during this pre-reform period was in an extensive way characterized by low efficiency of energy use (Zhang, 2005; Fei et al., 2011). Environmental protection was of seldom concern in this period, and the first law with possible effects on air quality protection was issued only after 1978 in China, which gradually increased to 15 in 1989 (Fig. 5B). This indicates a beginning, even if not so efficient, of pollution control. Therefore, the strongest decline in sunshine duration was observed in this period and a large difference can be noted in urban and rural sunshine duration trends, which declined by 0.20 h d$^{-1}$ decade$^{-1}$ and 0.14 h d$^{-1}$ decade$^{-1}$ respectively for 1960–1989 (Fig. 4A). Meanwhile, urban and rural total cloud cover slightly decreased by 0.41% decade$^{-1}$ and 0.48% decade$^{-1}$ respectively (Fig. 4B), which thus cannot explain the decrease in sunshine duration and the obvious difference in urban and rural dimming."

2. The authors argued that urbanization might not be able to reflect variation of atmospheric environment sine 1990s because regulations were gradually taken into action. This is somewhat speculative and needs some evidence. Satellite aerosol data such as MODIS AOD products during last 16 years may be used to shed light on this issue, otherwise, the speculation is not acceptable.

Reply 2: Thanks very much for this valuable comment. We will add a discussion based on recent satellite aerosol trends in China in the revised version of the manuscript. Based on your major comments 1 and 2, the fourth paragraph of Section 3.2 will be thoroughly modified as follows:

"In the brightening phase between 1990 and 2013, urban population sharply increased by 196 million persons decade$^{-1}$ and its proportion reached 54% in the end, indicating that urban population exceeded rural population in China (Fig. 4C). Besides, the primary industry only accounted for ~ 10% of total GDP in the year 2013, further indicating a more urbanized status. However, the increasing urbanization process did not result in a stronger decrease in sunshine duration but in contrast in a levelling off by –0.01 h d$^{-1}$ decade$^{-1}$ in both rural and urban areas (Fig. 4A). This suggests an insignificant urbanization effect on sunshine duration in the

brightening phase. Meanwhile, the total cloud cover trend in urban and rural areas significantly recovered by 1.23% decade$^{-1}$ and 1.03% decade$^{-1}$ respectively (Fig. 4B), thus not contributing to the slow-down of the declining trend in sunshine duration. However, effective air pollution regulations after 1990 are indicated in Fig. 5B, in that the number of air pollution-related laws and regulations rapidly increased to 135 in the year of 2013. The investment completed in the treatment of environmental pollution in China in 2013 (952 billion yuan) was 9.4 times of that in 2000 (101 billion yuan) (Fig. 5C). A slow-down in the increasing trend of total population to a rate of 92 million persons decade$^{-1}$ was also noted for the period of 1990–2013 when the one-child policy was implementing in China (Fig. 4C). In addition, the tertiary industry, which contributes less to air pollution than the secondary industry, kept growing in China during 1990–2013 and contributed equally as the secondary industry to the national economy in the end (Fig. 5A). The national pollution control efforts may have helped to offset anthropogenic air pollution induced during the urbanization process in the brightening phase, so that the trend of aerosol optical depth (AOD) simulated by the GOCART global chemical transport model started to decrease after 1996 (Streets et al., 2008). Consistently, a decline in PM$_{2.5}$ and PM$_{10}$ concentrations was noted in the 2000s (Lei et al., 2011; Wang et al., 2012a; Wang et al., 2013). However, using TOMS AOD products (1980–2001) along with MODIS/Terra AOD data (2000–2008), Guo et al. (2011) observed a continuous upward trend in average AOD (550 nm) over eight typical regions across China without any transition in 1990. Nevertheless, a significant increment of aerosol single scattering albedo was reported in China, which could result in less absorption and thus more radiation reaching the Earth's surface (Qian et al., 2007). The difference in the trends between PM concentrations and satellite AOD might be explained by the emission-control policies in China, which target primary aerosols (mainly related to PM) but are less strict for secondary aerosol precursors (e.g., NO$_x$, NMVOC and NH$_3$, mainly related to AOD) (Lin et al., 2010). In general, in the dimming phase without effective pollution regulations, the emissions generated during the urbanization process were directly changed into equivalent pollutants. On the other hand in the brightening phase, the increasing emissions were compensated by the clean air policies and investments, thus urbanization no longer simply meant an increase in air pollution and its effect on sunshine duration variations became insignificant."

Please note that according to the comments of Dr. Tanaka, the number of laws and regulations in Fig. 5B will be changed. Only those having possible effects on air quality protection will be picked out in the revised version. After this modification, the general trend of the accumulated number of laws and regulations remains similar to the old version.

References:

Guo, J. P., Zhang, X. Y., Wu, Y. R., Zhaxi, Y. Z., Che, H. Z., La, B., Wang, W., and Li, X. W.: Spatio-temporal variation trends of satellite-based aerosol optical depth in China during 1980–2008, Atmospheric Environment, 45, 6802-6811, 10.1016/j.atmosenv.2011.03.068, 2011.
Lin, J., Nielsen, C. P., Zhao, Y., Lei, Y., Liu, Y., and McElroy, M. B.: Recent changes in particulate air pollution over China observed from space and the ground: Effectiveness of emission control, Environ. Sci. Technol., 44, 7771-7776, 10.1021/es101094t, 2010.

Qian, Y., Wang, W., Leung, L. R., and Kaiser, D. P.: Variability of solar radiation under cloud-free skies in China: The role of aerosols, Geophys. Res. Lett., 34, L12804, doi:10.1029/2006gl028800, 2007.
Streets, D. G., Yu, C., Wu, Y., Chin, M., Zhao, Z., Hayasaka, T., and Shi, G.: Aerosol trends over China, 1980–2000, Atmospheric Research, 88, 174-182, 10.1016/j.atmosres.2007.10.016, 2008.

3. The authors used sunshine duration as a proxy data for surface solar radiation. It should be noted that sunshine duration mainly reflect cloud information. It may be argued that long-term variation of cloud cover did not support that of sunshine duration. However, cloud cover can not reflect all effects of cloud on surface solar radiation, furthermore, long-term trend of cloud cover obtained from surface manual observations is not free of large uncertainty. Therefore, it is suggest to exclude cloud effect on surface solar radiation in the analysis, for example, to perform similar analysis based on sunshine duration measurements in the case of cloud cover of 20%.

Reply 3: We appreciate your suggestion and agree with you that cloud effects on sunshine duration variations are not negligible. However, this study mainly focuses on the difference in sunshine duration variations between urban and rural areas. The selected urban-rural pairs are within 1°×1°. Regional cloud effects should therefore not cause the local difference in sunshine duration variations. This is also indicated in Fig. 4B, since urban and rural cloud trends almost coincide.

Since clouds have insignificant effects on the difference between urban and rural sunshine duration variations, it is conceivable that applying this study under clear-sky conditions will lead to the same conclusions. In the dimming phase, after excluding the effect of slightly decreasing total cloud cover, the decline in both urban and rural sunshine duration might be strengthened, which is also mentioned in previous studies under cloud-free skies (Qian et al., 2006, 2007). This may further verify a significant urbanization effect on sunshine duration in the dimming phase. Since the difference in urban and rural total cloud cover trends is very small (only 0.07% decade$^{-1}$), the ratio of rural to urban dimming might remain the same after excluding cloud effects. In the brightening phase, after excluding the effect of increasing total cloud cover, even an increasing trend in sunshine duration can be expected. This can be inferred from the stronger brightening trend in solar radiation in China estimated under clear-sky conditions (4 W m$^{-2}$ decade$^{-1}$, Wang et al., 2014) than under all-sky conditions (2.7 W m$^{-2}$ decade$^{-1}$, Shi et al., 2008). This would indicate an insignificant urbanization effect on sunshine duration variations in the brightening phase, similar with the conclusion made from all-sky conditions.

We agree with the reviewer that visually observed total cloud cover contains large uncertainties due to subjective eye observations and cannot fully represent all cloud effects. We will add this as a limitation at the end of the last paragraph of Section 3.2 as flows:

"The reason for the recent levelling off in the sunshine duration trend in China is still an open question and needs further discussion. Besides cloud cover, which contains uncertainties due to subjective eye observations, the cloud type as well as physical and radiative properties should also be fully considered. Effects from other potential driving factors, such as surface albedo, water vapor and wind speed, need to be clarified".

Finally, we would like to highlight that sunshine duration records can contain signals of the effects of aerosols, as reviewed by Sanchez-Romero et al. (2014).

References:

Qian, Y., Kaiser, D. P., Leung, L. R., and Xu, M.: More frequent cloud-free sky and less surface solar radiation in China from 1955 to 2000, Geophys. Res. Lett., 33, L01812, doi:10.1029/2005gl024586, 2006.
Qian, Y., Wang, W., Leung, L. R., and Kaiser, D. P.: Variability of solar radiation under cloud-free skies in China: The role of aerosols, Geophys. Res. Lett., 34, L12804, doi:10.1029/2006gl028800, 2007.
Shi, G., Hayasaka, T., Ohmura, A., Chen, Z., Wang, B., Zhao, J., Che, H., and Xu, L.: Data quality assessment and the long-term trend of ground solar radiation in China, Anglais, 47, 1006-1016, 2008.
Sanchez-Romero, A., Sanchez-Lorenzo, A., Calbó, J., González, J. A., and Azorin-Molina, C.: The signal of aerosol-induced changes in sunshine duration records: A review of the evidence, J. Geophys. Res.-Atmos., 119, 2013JD021393, doi:10.1002/2013JD021393, 2014.
Wang, Y., Yang, Y., Zhou, X., Zhao, N., and Zhang, J.: Air pollution is pushing wind speed into a regulator of surface solar irradiance in China, Environmental Research Letters, 9, 054004, 2014.
Wang, Y. W., Yang, Y. H., Han, S. M., Wang, Q. X., and Zhang, J. H.: Sunshine dimming and brightening in Chinese cities (1955-2011) was driven by air pollution rather than clouds, Climate Research, 56, 11-20, doi:10.3354/cr01139, 2013.

4. There are some stations showing long-term trend of sunshine duration quite different from that obtained in the majority of stations (Figure 2), some discussion on this issue is required. This might be related to cloud variation since the general tendency in the urbanization in China should be not quite different.

Reply 4: Good suggestion! We will add a discussion on the exceptional regions after line 7, Page 5 (1st paragraph of Section 3.1) as follows:

"The main regions that deviate from the dimming trend in both urban and rural areas are the Qinghai-Tibetan Plateau and Northeastern China (Fig. 2A and 2B), where water vapor and deep cloud cover were identified as critical regulators (Wang et al., 2011; Yang et al., 2012)."

References:

Wang, C. H., Zhang, Z. F., and Tian, W. S.: Factors affecting the surface radiation trends over China between 1960 and 2000, Atmos. Environ., 45, 2379-2385, doi:10.1016/j.atmosenv.2011.02.028, 2011.
Yang, K., Ding, B., Qin, J., Tang, W., Lu, N., and Lin, C.: Can aerosol loading explain the solar dimming over the Tibetan Plateau?, Geophys. Res. Lett., 39, L20710, doi:10.1029/2012GL053733, 2012.

5. The manuscript should be polished in language.

Reply 5: We will make a great effort to improve the English writing of this paper. The language issues mentioned in your minor comments will be carefully revised.

**Minor comments**

1. Abstract, this ratio should be defined.

Reply 1: We will add the definition as "The ratio of rural to urban dimming" in line 19, Page 1.

2. Why the urbanization effect on dimming diminished when urbanization level exceeds 50%. I wonder whether there are regions where the urbanization level reached to this level in the eastern China before 1990s.

Reply 2: Sorry for this confusing statement. The urbanization indices of *ULE*, *UP* and *PD* were calculated based on the data for 2013, to reflect the general condition of each province for the past decades for comparisons. We will correct the sentence in lines 19-21, Page 1 as follows:

"The ratio of rural to urban dimming generally increases from a minimum of 0.39 to a maximum of 0.87 with increasing indices of urbanization calculated based on the year of 2013, reaching saturation when the urbanization level exceeds 50%, or the urban population exceeds 20 million persons, or the population density becomes higher than 250 person km$^{-2}$."

Accordingly, the statement in the Conclusion section on line 1, Page 9 will be modified as:

"The ratio overall changes from the minimum of 0.39 to the maximum of 0.87 under different conditions of urbanization calculated based on the data for 2013."

3. P2, L24, delete due to the nature of their composition.

Reply 3: We will delete it.

4. P2, L35, delete furthermore

Reply 4: We will delete it.

5.P3, L3, since sunshine duration is related to cloudiness, why is it excluded in the analysis

Reply 5:  Please refer to the reply to your major comment 3.

6. P3, L10, urbanization also occurs in the rural regions?

Reply 6: We will modify the last paragraph of the Introduction section into:

"This study will then make a first attempt to examine the urbanization effect on sunshine duration variations in China. The wide temporal and spatial coverage of sunshine duration observations in China provides a unique opportunity to fully understand the differences of the dimming and brightening phenomenon between rural and urban areas. The value of urbanization as an indicator of pollution level will be evaluated for the dimming and brightening phases, respectively. In conclusion, the effect of urbanization on sunshine duration variations will be quantified."

7. P4, L20-25, what's difference between county and county-level cities

Reply 7: Most county-level cites in China were created during the 1980s-1990s by replacing counties. Compared to counties, county-level cities have less rural population, more developed economy, and better equipped public infrastructure. Counties are mainly governed by prefecture-level divisions, while county-level cities are usually governed by province-level divisions.

8. P4, L3-5, i do not understand why three big cities were excluded in the analysis since they may be use good examples for showing urbanization effect on surface solar radiation. This, certainly, is related to the method, I do not understand why the analysis were performed in each province.

Reply 8: We understand your concern. That was not an easy decision for us to exclude these three municipalities. But unfortunately, there were no available or suitable rural stations nearby to compare with the three municipalities. We applied two standards to choose the urban-rural pairs. The first is depending on the administrative divisions. There was no available rural station with long-term records directly under the jurisdiction of the three municipalities. The second is the location within 1° × 1° and of similar elevation. This second request seemed feasible also for the three municipalities. But in the other cases, the selected rural-urban pairs were in the same province and so were characterized by a similar background of urbanization process and economic development. By contrast, selecting a rural station for the three municipalities requires to find one in their neighboring provinces. Using a rural station in Hebei province (the neighboring province of Beijing, with the urbanization level of 48% in 2013) to compare with the urban station in Beijing (urbanization level of 86% in 2013) is not suitable. Taking into account these limitations, excluding the three municipalities was in our opinion the best choice.

Besides the reasons mentioned above, another reason for performing the analysis on provincial scale is because the official statistics on urbanization in China were based on this administrative unit.

9. P4, L7-8, not understood.

Reply 9: Sorry for the confusing statement. We will correct the sentence as "Records from 19 of the selected stations (~5.5% of total) were completed based upon their collocated stations with similar climatic and administrative conditions."

10. Annual ULE, PD, USP data?

Reply 10: The reviewer is right. We will replace "urbanization level, population density, and urbanization speed" with "annual *ULE*, *PD*, and *USP*" in the revised version of the manuscript.

11. Why the trends derived in rural and urban stations did not overlap in 14% pairs.

Reply 11: The urbanization impact has two aspects: (1) it describes the general increase in air pollution emissions which leads to a decrease in sunshine duration in both urban and rural areas, and (2) may lead to a difference between rural and urban sunshine duration trends. The difference in 14% of the pairs confirms that there are differences in the trend observed in urban and rural stations without negating the finding of a nationwide dimming.

12. P5, L 16-20, Since sunshine duration remains to decrease slightly in China, it seems not suitable to say brightening.

Reply 12: Based on your suggestion, the 2$^{nd}$ paragraph of Section 3.1 will be modified into:

"Unlike in the dimming phase where the decline in sunshine duration is almost nationwide distributed, China's sunshine duration trends in the brightening phase depict a more complex spatial pattern. From Fig. 2C and 2D it can be inferred that in the brightening phase, an increase in sunshine duration occurs in about half of China, especially in the regions of southern, northwestern and northeastern China, while dimming continues in the other half, especially in the North China Plain, where haze pollution is being reported (Wang et al., 2014c). This spatial pattern of sunshine duration trends is obtained in both rural and urban areas, suggesting a regional rather than local phenomenon. About 84% of the rural and urban sunshine duration changing rates overlap in the brightening phase (Fig. 3B). Averaged over the whole China, the sunshine duration trend in both urban and rural stations levels off to −0.01 h d$^{-1}$ decade$^{-1}$ for 1990–2013 (Fig. 4A)."

13. P6, L3-5, total cloud cover stabilized or decreased?

Reply 13: Sorry for causing this misunderstanding. We will correct this sentence in the revised manuscript as follows:

"Meanwhile, urban and rural total cloud cover slightly decreased by 0.41% decade$^{-1}$ and 0.48% decade$^{-1}$ respectively (Fig. 4B), which thus cannot explain the decrease in sunshine duration and the obvious difference in urban and rural dimming."

14. P6, L26-30, it should be cloud cover effect, not cloud effect. I do not agree with this discussion on cloud effect.

Reply 14: Thanks for the correction. Sorry for the limitation of this study, which will be stated in the revised version. We still deem that clouds should not be the main cause of the local difference in sunshine duration trends in China. Please refer to the reply to your major comment 3.

15. P7, L5-10, It seems in the Wang et al. study, rural-urban pairs are discussed in the 5*5 degree grid.

Reply 15: Thank you for raising this point. This information was not mentioned in their paper. Since the number of urban and rural stations in their study were not equal and evenly spread, it's also difficult to make an estimation from their Fig. 1.

---

## Referee Comment (RC3) · Anonymous Referee #1 · 8 Nov 2016

This paper uses the sunshine duration data from 1960 to 2013 over 344 stations in China to analyze (1) if the solar "dimming" and "brightening" is global or local, and (2) the urbanization effects on the "dimming" and "brightening" trends. While this paper reports some interesting features and includes large amount of data and auxiliary information, there are several fundamental shortcomings of this work, including the methods and the conclusions. My major concerns are listed below.

1. Sunshine duration and dimming/brightening: These terms are used interchangeably sometimes in the paper and yet they are differentiated other times, which causes confusing. For one thing, the sunshine duration is mostly determined by the cloud cover. If the authors want to use the sunshine duration as a surrogate for downwelling solar

radiation reaching the surface, they should at least stratify the sunshine duration according to the cloud cover, e.g., at cloud fraction <5%, <10%, or 20%, in order to get information on the role of pollution and clouds on sunshine duration. Without this first step, the results and conclusions about the urbanization effect on sunshine duration, which is a center focus of this paper, cannot be substantiated.

2. Urban and rural pair: Why is pairing necessary? It makes no sense to me to come up with those pairs. What is the criterion for paring? Distance, or wind direction? Many of them are very close with each other and it is impossible to link one rural site to a particular urban site to form a meaningful pair. Also, I cannot find any advantage of using the pairs in the analysis presented in the paper. It would make more sense to just categorize the available sites into "urban" and "rural" instead of pairing them. In this way, you don't have to exclude any sites due to the lack of "partner", such as Beijing, Shanghai, and Tianjin.

3. Urban or rural: It is not clear how the sites are defined; considering the fast urbanization level shown in Fig. 5 from <20% in 1960 to >50% in 2013, is there any rural sites in the 1960s became urban sites in later years? How is the information on urbanization level, urbanization speed, urban population, and population density used consistently at the places where these indicators are showing different indications? For example, Fig. 7 shows that the urbanization level is among the highest but the urbanization speed, population, and population density are among the lowest in the nation in the most northeast province (Heilongjiang)?

4. Urbanization vs. cloud cover change: The authors concluded that the trends of sunshine duration cannot be explained by the change of cloud cover, since both sunshine duration and cloud cover show decreasing trends before 1990, and after 1990 sunshine duration does not seem to change but cloud cover has increased. Rather, they attribute the sunshine duration trends before and after 1990 to urbanization and environmental regulation. This conclusion has several problems. First, as I suggested above in comment 1, the analysis should be stratified with cloud fraction. Second, Fig.

4 clearly shows anti-correlation between the interannual variation of sunshine duration and cloud cover: less cloud fraction, more sunshine duration, even though the trends do not seem to be consistent. Third, the urbanization levels have three distinct time periods according to Fig. 5: 1960-1978 (almost no change), 1978-1995 (moderate increase), and 1995-2013 (faster increase), which does not correspond to the sunshine duration trends, especially before 1978.

5. Dimming and brightening time periods: This is another confusing point. On the one hand, the authors claim the two periods of 1961-1989 and 1990-2013 that are "global dimming and brightening" periods, sounded like they are determined by the trends of downwelling solar radiation at the surface globally; on the other hand, it is stated in the paper that the periods are defined by the change point of sunshine duration trends in China. So, these two periods should not be called dimming or brightening periods, but the sunshine duration decreasing and no change periods. However, it is clear from Fig. 2 and also stated in the paper that half of the country is continuously showing the decrease of sunshine duration in the second period of 1990-2013, which indicate that the apparent "stable" trend of sunshine duration in 1990-2013 is not really stable but misleading, since is a result of averaging the positive and negative trends at different places.

6. Global or local: This argument is pointless. The authors use the sunshine duration trends over paired "urban" and "rural" sites in China to determine if the dimming/brightening are global or local phenomenon: If the paired sites are showing different trends then it is "large scale" (note: the authors do not call it "global"), but if the trends are different it could be considered as "local". This is kind of meaningless, because a) the "local" spatial scale is not defined, and b) even if all sites have the same trends, it still does not mean global, since all sites are located in China. This paper should just focus on the trends from >50 years of data, rather than discuss if the trends are global or local.

7. Urbanization and pollution emissions: There are many emission datasets available

for the study period. It would be illuminating to see how the sunshine duration trend at low cloud cover conditions corresponds to the emission trends of pollutants.

Overall, it requires major revision of the current paper. The analysis and methods should be redesigned to get meaningful results and conclusions.

Specific comments:

Page 1, line 14: Not clear how the dimming and brightening phases are determined.

Page 1, line 16: I am not aware of anyone had suggested urban dimming/rural brightening.

Page 2, line 27-28: You should at least check if AOD and API available in recent years are consistent with the sunshine duration, so you may use the sunshine duration as a proxy for pollution levels.

Page 3, line 2: "inhomogeneity" of what?

Page 3, line 7-8, objective (2): so the dimming and brightening are "global"? How do they relate to sunshine duration?

Page 3, line 16-17: Do you define the two periods according the sunshine duration trends? How do you explain the different trends in different areas (Fig. 2)?

Page 4, line 7: difference of 0 deg N and 0 deg E means the rural and urban stations are in the exact same locations? Why do you want to replace 19 stations? This is confusing. Also, as I mentioned earlier, you have not explain why selecting pairs are necessary.

Page 5, line 4-5: On page 3 the authors stated that the dimming and brightening periods were determined "according to the change-point in the early 1990s in sunshine duration trends in China evidenced by recent studies", but it is said here that the time periods were "GLOBAL dimming and brightening" periods! It is confusing and it should really be clarified how the two time periods were determined and why.

[Figure]

Page 5, line 9, large scale or local phenomenon: Need to be more specific about the scale – is provincial scale considered as local or large scale? I suggest use "...a wide spread phenomenon..." to be more appropriate.

Page 5, line 17-20: If only half of China is brightening while the other half is still dimming, why do you decide this is the phase of brightening, not dimming?

Page 5, line 25-27: But you just said a few line above that half of China is still dimming! Does it mean that the transition from dimming to brightening is a local scale phenomenon, not a countrywide phenomenon?

Page 5, line 30: Repeating the question I had earlier: does urbanization convert any rural sites to urban sites? Does the fixed urban/rural sites make sense for the entire 54 years?

Page 6, line 4-5 and 14: "stabilize" or "stable" are not used properly. Do you mean constant or unchanged?

Page 6, line 7: explain what "pre-reform" period is.

Page 6, line 12-13: What kind of environmental protection laws? How many of them are related to pollutant emission regulations that are relevant here? How effective are they? Why the regulations work differently in China as half of the country showing the increase of sunshine duration but another half continuously showing the decrease? Can you use the pollution emission datasets to corroborate with the effectiveness of the law?

Page 6, line 14: I don't understand the argument here – why is the strongest decline of sunshine duration in the 1980s when the environmental regulations started being implemented?

Page 6, line 16: clarify what you mean by "transition from extensive to intensive".

Page 6, line 26: There is not enough evidence to support the conclusion of urbanization

effect on sunshine duration. Showing their respective trends does not mean causal relationship.

Page 6, line 31: Where is this "widening urban-rural contrast" shown? Fig 4A does not seem to suggest a "widening" trend.

Page 7, line 10, first sentence: not substantiated.

Page 7, line 16-17: I cannot see these relationships between Fig 2 and 7. For example, the dimming/brightening trends In the NW province (Xinjiang) have the same size as the trends in the eastern coastal provinces shown in Fig 2, but the urbanization level, speed, population, and population density in Fig 7 are very different between Xinjiang and eastern coastal provinces.

Page 8, line 14-15, (1): no air pollution emissions are shown in the paper.

Page 8, line 15, (2): is this a conclusion?

Page 8, line 19: clarify what "a large overlap" is.

Page 8, line 10-21: How big is the region to see the "regional phenomenon"? Certainly it is not the entire China.

Page 8, line 28-29: "As a consequence..." Not true. The apparent insignificant trend is the consequence of averaging half positives and half negative trends together to make the overall mean trend flat. There is no solid analysis provided in this paper showing the effectiveness of the air pollution control on the sunshine duration.

---

## Author Comment (AC3) · 5 Dec 2016

**Reply to Anonymous Referee #1**

Thanks very much for taking valuable time and energy in reviewing this manuscript. Please find the point-by-point responses to your specific comments in the following context.

**Major comments**

1. Sunshine duration and dimming/brightening: These terms are used interchangeably sometimes in the paper and yet they are differentiated other times, which causes confusing. For one thing, the sunshine duration is mostly determined by the cloud cover. If the authors want to use the sunshine duration as a surrogate for downwelling solar radiation reaching the surface, they should at least stratify the sunshine duration according to the cloud cover, e.g., at cloud fraction <5%, <10%, or 20%, in order to get information on the role of pollution and clouds on sunshine duration. Without this first step, the results and conclusions about the urbanization effect on sunshine duration, which is a center focus of this paper, cannot be substantiated.

Reply 1: Based on your comments, the whole manuscript will be thoroughly revised to standardize the usage of the terms of sunshine duration and dimming/brightening. We also agree with you that under all-sky conditions the cloud effect has a major influence on sunshine duration variability, and thus it is necessary to remove it in order to fully understand and discuss the aerosol effect. However, regional cloud effects should not play a major role in the main findings of this study (rural vs urban trends in sunshine duration over China) as indicated in Fig. 4B where urban and rural cloud trends almost coincide. Since clouds have non-significant effects on the local difference in sunshine duration variations (the difference between urban and rural total cloud cover trends is only 0.07% decade$^{-1}$), it is conceivable that stratifying the analysis with cloud fraction will lead to the same conclusions. Therefore, an exclusion of cloud effects is not necessary in our opinion for understanding the differences in urban and rural sunshine duration trends.

In addition, we are studying urban effects on sunshine duration series, which can include both direct and indirect aerosol effects. Thus, the goal is not to determine if direct aerosol effects or indirect aerosol-cloud interactions are the causes of the potential disagreements (if observed).

2. Urban and rural pair: Why is pairing necessary? It makes no sense to me to come up with those pairs. What is the criterion for paring? Distance, or wind direction? Many of them are very close with each other and it is impossible to link one rural site to a particular urban site to form a meaningful pair. Also, I cannot find any advantage of using the pairs in the analysis presented in the paper. It would make more sense to just categorize the available sites into "urban" and "rural" instead of pairing them. In this way, you don't have to exclude any sites due to the lack of "partner", such as Beijing, Shanghai, and Tianjin.

Reply 2: Just categorizing available sites into "urban" and "rural" without pairing them may include significant spurious biases. In this case, the categorized urban and rural stations may

face with very different conditions in terms of climate, geographical location, orography, land use, etc. These factors may also induce differences in the sunshine duration trends between the categorized stations, and thus limit the understanding of urbanization effect. Therefore it is necessary to restrict the above mentioned differences by using the pairing method to allow for an appropriate comparison of rural and urban trends under similar conditions, which has been demonstrated by previous studies to be a valid method and frequently used in the context of the urban effect on temperature trends (e.g., Hausfather et al., 2013; Wang et al., 2014a).

In this study, two criteria were given for selecting the urban-rural pairs: (1) the administrative divisions (i.e., the selected county is under the jurisdiction of the corresponding city) and/or (2) location (i.e., the urban-rural station pairs are within 1° × 1° and of similar elevation). We added the administrative division criterion to further ensure a similar geographical and developmental background for the rural and urban stations, since the administrative division in China has fully considered the principles of the politics, economics, ethnics, demographics, historical traditions, topography and geomorphology, etc. In the revised manuscript, we will add a sentence to explain better the advantage of the pairing method and the criteria used to define the pairs. According to your comments, the third paragraph of Section 2 will be modified into:

"The raw dataset encompasses 906 stations across latitudes 16°32′–52°58′ N, longitudes 75°14′–132°58′ E, and elevations 2–4800 m. Only stations with sunshine duration records covering at least 85% (≥ 45 years) of the period 1960–2013 were considered. The pairing method was then applied to yield a set of proximate urban/rural station pairs that should be relatively unaffected by biases introduced by differences in the prevailing climate regime, geographical location, etc. In this way, potential differences in the trends can be more directly related to urbanization effects. The urban-rural station pairs were strictly picked out for each of the 28 studied provincial-level divisions across China, including 22 provinces across the Mainland China, 5 autonomous regions, and 1 municipality (Chongqing) (Fig. 1), depending on (1) the administrative divisions (i.e., the selected county is under the jurisdiction of the corresponding city) and/or (2) location (i.e., the urban-rural station pairs are within 1° × 1° and of similar elevation). Here the criterion of administrative division was added to further ensure a similar geographical and developmental background for the selected urban-rural station pairs. The administrative division in China has fully considered the principles of the politics, economics, ethnics, demographics, historical traditions, topography, geomorphology, etc. Please note that the other three municipalities of Beijing, Shanghai and Tianjin, where only urban stations meet the criterion of having ≥ 45 years of available data, were excluded from this study due to a lack of corresponding rural stations. The pairing method thus ensures an appropriate comparison of rural and urban trends under similar conditions."

3. Urban or rural: It is not clear how the sites are defined; considering the fast urbanization level shown in Fig. 5 from <20% in 1960 to >50% in 2013, is there any rural sites in the 1960s became urban sites in later years? How is the information on urbanization level, urbanization speed, urban population, and population density used consistently at the places where these indicators are showing different indications? For example, Fig. 7 shows that the urbanization level is

among the highest but the urbanization speed, population, and population density are among the lowest in the nation in the most northeast province (Heilongjiang)?

Reply 3: We appreciate the reviewer's comments. Urban and rural stations in this study were defined directly based on the current administrative divisions of China. In most cases, the stations nowadays are in the same administrative level as in the 1960s. But the reviewer is right that there are cases of rural sites becoming urban sites in later years. Most of the cases correspond to counties becoming county-level cities or city-governed districts, which were regarded as urban stations in this study. We also understand your concern on the urbanization indices. A situation like Heilongjiang noted by the reviewer might happen. Heilongjiang is located in the most northeastern part of China with long and bitter winters. Therefore, this area is sparsely populated, which is reflected by the low urban population and population density. A high urbanization level in Heilongjiang indicates a high ratio of urban to total population, namely that more than 55% of the total population is urban population. And the urbanization level in Heilongjiang is relatively high even in the 1960s, so that the urbanization speed is among the lowest. We didn't expect these indices to be totally consistent and for this reason we used the different indices to fully represent urbanization and its effect on sunshine duration trends. As shown in Fig. 8, the ratio of rural to urban dimming generally shows a similar relationship with the different urbanization indices. Based on your comments, the second paragraph of Section 2 will be modified to further clarify the definition of urban and rural stations in this study as follows:

"Urban/rural stations were defined according to the administrative divisions of China. There are five administrative levels of local government in China: the provincial level (province, autonomous region, municipality, and special administrative region), the prefectural level (prefectural-level city, sub-provincial-level city, autonomous prefecture, prefecture, leagues, etc.), the county level (county-level city, city-governed district, autonomous county, county, banner, etc.), the township level (sub-district, county-level district, town, township, etc.) and the village level (neighbourhood, community, village, etc.) (http://www.gov.cn/). In this study, meteorological stations located in the provincial/prefectural-level divisions were defined as urban stations, while those located in county/township/village-level divisions were deemed as rural stations. Special cases are county-level cities and city-governed districts, which belong to county-level divisions but were considered as urban stations in this study for two reasons. First, a rapidly accelerating process of urbanization can be expected in county-level cities and city-governed districts, which are mainly created by replacing counties. Besides, county-level cities and city-governed districts are mainly located in/near to prefectural-level cities, and therefore might be frequently influenced by urban pollution or become the source of pollution emissions as a result of industry transfer."

4. Urbanization vs. cloud cover change: The authors concluded that the trends of sunshine duration cannot be explained by the change of cloud cover, since both sunshine duration and cloud cover show decreasing trends before 1990, and after 1990 sunshine duration does not seem to change but cloud cover has increased. Rather, they attribute the sunshine duration trends before and after 1990 to urbanization and environmental regulation. This conclusion has

several problems. First, as I suggested above in comment 1, the analysis should be stratified with cloud fraction. Second, Fig. 4 clearly shows anti-correlation between the interannual variation of sunshine duration and cloud cover: less cloud fraction, more sunshine duration, even though the trends do not seem to be consistent. Third, the urbanization levels have three distinct time periods according to Fig. 5: 1960-1978 (almost no change), 1978-1995 (moderate increase), and 1995-2013 (faster increase), which does not correspond to the sunshine duration trends, especially before 1978.

Reply 4: We understand your concern. For your first point, we still deem that clouds should not be the main cause of the local difference in sunshine duration trends in China, therefore an exclusion of cloud effects is not mandatory in understanding the differences in urban and rural sunshine duration trends. In fact, the main objective of the manuscript is not to explain the causes of sunshine duration trends, which have already been well discussed in previous studies, but rather to explain the differences between urban and rural sunshine duration trends. Please refer to the reply to your major comment #1.

For your second point, it has been proven that clouds play an important role in modulating solar radiation at daily to inter-annual timescales while aerosols become important for the variability of solar radiation at a decadal timescale in China (Xia, 2010; Wang et al., 2012a). Again, it is not the goal of our study to discuss the causes of the trends in sunshine duration in China, especially because there is a large literature for this topic. Nevertheless, we have modified the manuscript accordingly in order to avoid relating our trends in sunshine duration (both in rural and urban areas) only to aerosol changes.

For your third point, the inconsistency between urbanization level and sunshine duration trends before 1978 might be due to the official calculation method of urban and rural population. In the urban population only the formally registered urban residents were considered, while people who worked in urban areas without a registered urban residence were still counted as rural population. At that time and even now, it is not so easy for the migrant workers to get a registered residence from the cities they work in. This population calculation method was changed from registered to permanent residents during the 1980s. Since the calculation of urbanization level contains such issues, and Referee #2 has also raised concerns about this evolution, the bold line in Fig. 5 will be deleted, and accordingly Fig. 5 will be combined with Fig. 4 as Fig. 4C. An increasing urbanization process during the 1960–1978 period still can be inferred from the GDP composition trends shown in the figure below. This figure will replace Fig. 6A of the old version of the manuscript and become Fig. 5A in the revised manuscript (since the former Fig. 5 will become the part C of Fig. 4). The secondary sector of the economy increased by 7.6% decade$^{-1}$ during the 1960–1978 period, whereas the primary sector decreased by 3.3% decade$^{-1}$. This suggests a transfer from primary industry to secondary industry, which is one of the main characteristics of urbanization.

[Figure]

Accordingly, the second and third paragraph of Section 3.2 will be modified into:

"In the dimming phase between 1960 and 1989, China experienced a boom in population and industrialization. China's population rapidly increased by an almost linear trend of 166 million persons decade$^{-1}$ (Fig. 4C). The urban population only accounted for a small portion of the total population, and grew slowly during this period. One possible explanation to this is the calculation method of urban population in China before the 1980s, where only the formally registered urban residents were counted in. In this case, people working in urban areas but without a registered urban residence were still counted as rural population. This standard for calculating population was changed from registered to permanent residents during the 1980s. An increasing urbanization process during the dimming phase can be inferred also from the GDP composition trends (Fig. 5A). The portion of the secondary sector in China's GDP largely increased by 4.2% decade$^{-1}$ during the period of 1960–1989. Meanwhile, the primary sector decreased by 3.8% decade$^{-1}$. This suggests a transition from the primary to the secondary sector. The secondary sector, which contributed up to 48% to the total GDP, became the backbone of China's booming economy at this stage. The industrial growth during this period was in an extensive way characterized by low efficiency of energy use (Zhang, 2005; Fei et al., 2011). Environmental protection was of seldom concern in this period, and the first law with possible effects on air quality protection was issued only after 1978 in China, which gradually increased to 15 in 1989 (Fig. 5B). This indicates a beginning, even if not so efficient, of pollution control. Therefore, the strongest decline in sunshine duration was observed during 1960–1989 and a discernible difference can be noted in urban and rural sunshine duration trends, which declined by 0.20 h d$^{-1}$ decade$^{-1}$ and 0.14 h d$^{-1}$ decade$^{-1}$ respectively (Fig. 4A). Meanwhile, urban and rural total cloud cover slightly decreased by 0.41% decade$^{-1}$ and 0.48% decade$^{-1}$ respectively (Fig. 4B), which thus can explain neither the decrease in sunshine duration nor the obvious difference in urban and rural trends."

5. Dimming and brightening time periods: This is another confusing point. On the one hand, the authors claim the two periods of 1961-1989 and 1990-2013 that are "global dimming and brightening" periods, sounded like they are determined by the trends of downwelling solar radiation at the surface globally; on the other hand, it is stated in the paper that the periods are

defined by the change point of sunshine duration trends in China. So, these two periods should not be called dimming or brightening periods, but the sunshine duration decreasing and no change periods. However, it is clear from Fig. 2 and also stated in the paper that half of the country is continuously showing the decrease of sunshine duration in the second period of 1990-2013, which indicate that the apparent "stable" trend of sunshine duration in 1990-2013 is not really stable but misleading, since is a result of averaging the positive and negative trends at different places.

Reply 5: The reviewer is right. Sorry for causing the confusions. The word "global" used in the "global dimming and brightening" phenomenon was originally referred to global radiation rather than a global-scale dimension. This phenomenon has been widely discussed in specific regions/countries all over the world, e. g. the United States (Stanhill and Cohen, 2005), the Iberian Peninsula and Western Europe as a whole (Sanchez-Lorenzo et al. , 2007, 2008), Japan (Stanhill and Cohen, 2008), New Zealand (Liley, 2009), India (Jaswal, 2009), Italy (Manara et al., 2016), etc. In this study, we used the "dimming" and "brightening" phases to enable the comparisons of this phenomenon in China with other regions/countries as well as the global trend. For the global trend, a transition was noted since the late 1980s (Wild et al., 2005). Consistently, a transition in 1990 was proven to occur not only in sunshine duration trend but also in surface solar radiation trend in China (Shi et al., 2008; Tang et al., 2011; Wang and Yang, 2014; Wang and Wild, 2016). We agree with you that the spatial pattern of sunshine duration variations since the 1990s is complicated in China and needs further analysis. A similar discussion will be added at the end of the last paragraph of Section 3.2 based on your comment. To avoid the misunderstanding pointed out in both your major comments 1 and 5, the term of "brightening phase" will be replaced by "levelling off phase", and the definition of the dimming and levelling off phases will be further clarified at the end of the first paragraph of Section 2 as follows:

 "The dimming and levelling off phases were defined as the periods of 1960–1989 and 1990–2013 respectively, according to the change-point in the early 1990s in sunshine duration trends in China evidenced by previous studies (e.g., Xia, 2010; Wang et al., 2013; Wang and Yang, 2014). The year 1990 was also identified as the transition year for the surface solar radiation trend in China (Shi et al., 2008; Tang et al., 2011). This is generally consistent with solar dimming and brightening phases coined at the global scale (Wild, 2009, 2012)."

6. Global or local: This argument is pointless. The authors use the sunshine duration trends over paired "urban" and "rural" sites in China to determine if the dimming/brightening are global or local phenomenon: If the paired sites are showing different trends then it is "large scale" (note: the authors do not call it "global"), but if the trends are different it could be considered as "local". This is kind of meaningless, because a) the "local" spatial scale is not defined, and b) even if all sites have the same trends, it still does not mean global, since all sites are located in China. This paper should just focus on the trends from >50 years of data, rather than discuss if the trends are global or local.

Reply 6: There might be some misunderstandings on the discussion of global or local phenomenon here. As already mentioned in the reply to your major comment #5, after the

global dimming and brightening phenomenon was put forward, "global", originally referring to "global radiation", was often interpreted in term of a global-scale dimension. Therefore, Alpert et al. (2005, 2008) raised an argument on whether the dimming was a global signal or just a local/urban signal. This hypothesis was given based on the highlighted role of anthropogenic aerosols on solar radiation variations. In their studies, the dimming phenomenon was supposed to be limited to urban/highly populated sites and thus a "local" phenomenon. Using a comprehensive set of sunshine duration records, the present study aims to investigate the same issue in China. The "local" spatial scale was clearly defined by selecting 172 urban and rural station pairs, shown in Fig. 2. In this way, it is clear to check whether the decadal sunshine duration variations only occurred in urban/rural regions or in both, namely whether they state a local or regional/national phenomenon. Similarly, evidences were given in New Zealand (Liley, 2009), Egypt (Robaa, 2009), Japan (Tanaka et al., 2016), etc. Anyway, the unnecessary use of "global" will be reduced to the least according to your comments.

7. Urbanization and pollution emissions: There are many emission datasets available for the study period. It would be illuminating to see how the sunshine duration trend at low cloud cover conditions corresponds to the emission trends of pollutants.

Reply 7: Thanks for the suggestion. But we are afraid that this is already beyond the scope of this study. Its focus is on the difference of urban and rural sunshine duration trends rather than how sunshine duration reacts to pollution, which has already been well discussed in previous works, please refer to Xia (2010), Wang et al. (2012b), etc. In the review of Sanchez-Romero et al. (2014), a comprehensive discussion has also been given on this issue. In addition, emission datasets as well as measurements on pollution indices are not adequately available especially in the dimming phase. This is one of the reasons for identifying an indicator for pollution like urbanization to investigate the role of aerosols in sunshine duration/solar radiation variations.

**Minor comments**

1. Page 1, line 14: Not clear how the dimming and brightening phases are determined.

Reply 1: Please refer to the reply to your comment #5.

2. Page 1, line 16: I am not aware of anyone had suggested urban dimming/rural brightening.

Reply 2: According to your comment, "urban dimming or rural brightening" will be changed into "local effects".

3. Page 2, line 27-28: You should at least check if AOD and API available in recent years are consistent with the sunshine duration, so you may use the sunshine duration as a proxy for pollution levels.

Reply 3: Sorry for this misunderstanding. But here the proxy for pollution levels referred to urbanization rather than sunshine duration.

4. Page 3, line 2: "inhomogeneity" of what?

Reply 4: According to your comment, "inhomogeneity" will be clarified as follows:

"Sunshine duration is used as the proxy for surface solar radiation. Compared with surface solar radiation, sunshine duration has a much wider spatial and temporal coverage and is almost free of temporal inhomogeneities in China (Xia, 2010; Wang and Yang, 2014; Wang et al., 2015). For the whole China (latitudes 16°32'–52°58' N and longitudes 75°14'–132°58' E), there are 130 solar radiation stations, with only 59 stations covering at least 85% of the measurement period from 1958 to the present. Moreover, the majority of surface solar radiation stations are located in urban areas, which will further limit the available data samples for studying urbanization effects. In contrast, there are 906 sunshine duration stations, thus exceeding the solar radiation stations by a factor of about 7. Besides, 638 of them have sunshine duration records which cover at least 85% of the whole measurement period since the 1950s. Moreover, it has been demonstrated that sunshine duration is able to capture variations in cloudiness and the signal from aerosol concentrations (Sanchez-Lorenzo et al., 2009; Wang et al., 2012c; Sanchez-Romero et al., 2014, 2016; Li et al., 2016; Wild, 2016). Similar to the surface solar radiation trend, a transition from decreasing to levelling off was also noted in sunshine duration trend in China around 1990 (Wang et al., 2013, 2014). Nevertheless, the existence of an urbanization impact on the trend of sunshine duration remains unclear."

5. Page 3, line 7-8, objective (2): so the dimming and brightening are "global"? How do they relate to sunshine duration?

Reply 5: According to your comment, the last paragraph of the Introduction Section will be changed into:

"This study will thus make a first attempt to examine the urbanization effect on sunshine duration variations in China. The wide temporal and spatial coverage of sunshine duration observations in China provides a unique opportunity to fully understand the differences of sunshine duration trends between rural and urban areas since the 1960s. The value of urbanization as an indicator of pollution level will be evaluated for the sunshine dimming and levelling off phases, respectively. In conclusion, the effect of urbanization on sunshine duration variations will be quantified."

6. Page 3, line 16-17: Do you define the two periods according the sunshine duration trends? How do you explain the different trends in different areas (Fig. 2)?

Reply 6: Please refer to the reply to your comment #5.

7. Page 4, line 7: difference of 0 deg N and 0 deg E means the rural and urban stations are in the exact same locations? Why do you want to replace 19 stations? This is confusing. Also, as I mentioned earlier, you have not explain why selecting pairs are necessary.

Reply 7: Thanks for the comments. The beginning of the fourth paragraph of Section 2 will be modified into:

"Finally, 172 urban-rural station pairs all over China were selected with an average difference of 95 m in elevation and a negligible difference in average latitude (~0.001°N) and longitude (~0.002°E) (Fig. 1). Records from 19 of the selected stations (~5.5% of total) were completed based upon their collocated stations with similar climatic and administrative conditions."

About the reason for selecting urban and rural station pairs, please refer to the reply to your major comment #2.

8. Page 5, line 4-5: On page 3 the authors stated that the dimming and brightening periods were determined "according to the change-point in the early 1990s in sunshine duration trends in China evidenced by recent studies", but it is said here that the time periods were "GLOBAL dimming and brightening" periods! It is confusing and it should really be clarified how the two time periods were determined and why.

Reply 8: According to your comment, "global dimming and brightening periods" will be replaced by "the 1960–2013 period".

9. Page 5, line 9, large scale or local phenomenon: Need to be more specific about the scale – is provincial scale considered as local or large scale? I suggest use "…a wide spread phenomenon…" to be more appropriate.

Reply 9: Good suggestion.  We will use it at the appropriate locations in the text.

10. Page 5, line 17-20: If only half of China is brightening while the other half is still dimming, why do you decide this is the phase of brightening, not dimming?

Reply 10: According to your comment, "brightening phase" will be replaced by "levelling off phase". Please refer to the reply to your major comment #5.

11. Page 5, line 25-27: But you just said a few line above that half of China is still dimming! Does it mean that the transition from dimming to brightening is a local scale phenomenon, not a countrywide phenomenon?

Reply 11: The term "local" means mainly urban area with "local" pollution sources. Rural is more related to background pollution and thus rather "non-local". Please refer to the reply to your major comment #6.

12. Page 5, line 30: Repeating the question I had earlier: does urbanization convert any rural sites to urban sites? Does the fixed urban/rural sites make sense for the entire 54 years?

Reply 12: Thanks for highlighting this issue. Please refer to the reply to your major comment #3.

13. Page 6, line 4-5 and 14: "stabilize" or "stable" are not used properly. Do you mean constant or unchanged?

Reply 13: Thanks for the correction. After the revision according to your major comment #4, the two sentences as well as the usage of "stabilize" or "stable" will not exist anymore.

14. Page 6, line 7: explain what "pre-reform" period is.

Reply 14: The year 1978 is the reform year in China with the adoption of the open door policy. "Pre-reform" period is used to refer to the period before 1978. After the revision according to your major comment #4, the term "pre-reform" will not be used anymore.

15. Page 6, line 12-13: What kind of environmental protection laws? How many of them are related to pollutant emission regulations that are relevant here? How effective are they? Why the regulations work differently in China as half of the country showing the increase of sunshine duration but another half continuously showing the decrease? Can you use the pollution emission datasets to corroborate with the effectiveness of the law?

Reply 15: We agree with your comment. We will double check the information on all the environmental laws and regulations as also suggested by another referee (Dr. Tanaka). Only those having possible effects on air quality protection will be picked out in the revised version of the manuscript. Accordingly, Fig. 6B will be redrawn and the corresponding text will be revised. Unfortunately, we lack of enough information to explore the effectiveness of these laws. We will add this as a limitation at the end of the fourth paragraph of Section 3.2.

16. Page 6, line 14: I don't understand the argument here – why is the strongest decline of sunshine duration in the 1980s when the environmental regulations started being implemented?

Reply 16: After the revision according to your major comment #4, this statement will not exist anymore and the second and third paragraph of Section 3.2 will be modified into:

"In the dimming phase between 1960 and 1989, China experienced a boom in population and industrialization. China's population rapidly increased by an almost linear trend of 166 million persons decade$^{-1}$ (Fig. 4C). The urban population only accounted for a small portion of the total population, and grew slowly during this period. One possible explanation to this is the calculation method of urban population in China before the 1980s, where only the formally

registered urban residents were counted in. In this case, people working in urban areas but without a registered urban residence were still counted as rural population. This standard for calculating population was changed from registered to permanent residents during the 1980s. An increasing urbanization process during the dimming phase can be inferred also from the GDP composition trends (Fig. 5A). The portion of the secondary sector in China's GDP largely increased by 4.2% decade$^{-1}$ during the period of 1960–1989. Meanwhile, the primary sector decreased by 3.8% decade$^{-1}$. This suggests a transition from the primary to the secondary sector. The secondary sector, which contributed up to 48% to the total GDP, became the backbone of China's booming economy at this stage. The industrial growth during this period was in an extensive way characterized by low efficiency of energy use (Zhang, 2005; Fei et al., 2011). Environmental protection was of seldom concern in this period, and the first law with possible effects on air quality protection was issued only after 1978 in China, which gradually increased to 15 in 1989 (Fig. 5B). This indicates a beginning, even if not so efficient, of pollution control. Therefore, the strongest decline in sunshine duration was observed during 1960–1989 and a discernible difference can be noted in urban and rural sunshine duration trends, which declined by 0.20 h d$^{-1}$ decade$^{-1}$ and 0.14 h d$^{-1}$ decade$^{-1}$ respectively (Fig. 4A). Meanwhile, urban and rural total cloud cover slightly decreased by 0.41% decade$^{-1}$ and 0.48% decade$^{-1}$ respectively (Fig. 4B), which thus can explain neither the decrease in sunshine duration nor the obvious difference in urban and rural trends."

17. Page 6, line 16: clarify what you mean by "transition from extensive to intensive".

Reply 17: After the revision according to your major comment #4, this statement will not exist anymore and the fourth paragraph of Section 3.2 will be modified into:

"In the levelling off phase between 1990 and 2013, urban population sharply increased by 196 million persons decade$^{-1}$ and its proportion reached 54% in the end, indicating that urban population started to exceed rural population in China (Fig. 4C). Besides, the primary sector only accounted for ~10% of total GDP in the year 2013, further indicating a more urbanized status. However, the increasing urbanization process did not result in a stronger decrease in sunshine duration but in contrast in an overall levelling off by –0.01 h d$^{-1}$ decade$^{-1}$ in both rural and urban areas (Fig. 4A). This indicates an insignificant urbanization effect on sunshine duration in this period. Meanwhile, the total cloud cover trend in urban and rural areas significantly recovered by 1.23% decade$^{-1}$ and 1.03% decade$^{-1}$ respectively (Fig. 4B), thus not contributing to the slow-down of the declining trend in sunshine duration. However, effective air pollution regulations after 1990 are indicated in Fig. 5B, in that the number of air pollution-related laws and regulations rapidly increased to 135 in the year of 2013. The investment completed in the treatment of environmental pollution in China in 2013 (952 billion yuan) was 9.4 times of that in 2000 (101 billion yuan) (Fig. 5C). A slow-down in the increasing trend of total population to a rate of 92 million persons decade$^{-1}$ was also noted for the period of 1990–2013 when the one-child policy was implementing in China (Fig. 4C). In addition, the tertiary sector, which contributes less to air pollution than the secondary sector, kept growing in China during 1990–2013 and contributed equally as the secondary sector to the national economy in the end (Fig. 5A). The national pollution control efforts may have helped to offset anthropogenic air pollution

induced during the urbanization process since the 1990s, so that the trend of AOD simulated by the GOCART global chemical transport model started to decrease after 1996 (Streets et al., 2008). Consistently, a decline in $PM_{2.5}$ and $PM_{10}$ concentrations was noted in the 2000s (Lei et al., 2011; Wang et al., 2012a; Wang et al., 2013). However, using TOMS AOD products (1980–2001) along with MODIS/Terra AOD data (2000–2008), Guo et al. (2011) observed a continuous upward trend in average AOD (550 nm) over eight typical regions across China without any transition in 1990. Nevertheless, a significant increment of aerosol single scattering albedo was reported in China, which could result in less absorption and thus more radiation reaching the Earth's surface (Qian et al., 2007). The difference in the trends between PM concentrations and satellite AOD might be explained by the emission-control policies in China, which target primary aerosols (mainly related to PM) but are less strict for secondary aerosol precursors (e.g., $NO_x$, NMVOC and $NH_3$, mainly related to AOD) (Lin et al., 2010). In general, in the dimming phase without effective pollution regulations, the emissions generated during the urbanization process were directly changed into equivalent pollutants. On the other hand, in the subsequent levelling off phase, the increasing emissions were compensated by the clean air policies and investments, thus urbanization no longer simply meant an increase in air pollution and its effect on sunshine duration variations became insignificant."

18. Page 6, line 26: There is not enough evidence to support the conclusion of urbanization effect on sunshine duration. Showing their respective trends does not mean causal relationship.

Reply 18: Thanks for this point. Please check the revised paragraphs mentioned in the above replies to your minor comments #16 and #17. We believe that the revised version of the manuscript will be improved enough to give a solid conclusion.

19. Page 6, line 31: Where is this "widening urban-rural contrast" shown? Fig 4A does not seem to suggest a "widening" trend.

Reply 19: It can be noted in Fig. 4A that the mean values of sunshine duration in urban areas are generally larger than in rural areas in the 1960s, while this situation converses in the 1980s. The widening contrast in the dimming phase has been somehow concealed because of the different start of the urban and rural lines.

Reply 20: Sorry for this. Please refer to the replies to your major comment #4 and minor comments #16 and #17. We believe this statement will be substantiated in the revised version of the manuscript.

21. Page 7, line 16-17: I cannot see these relationships between Fig 2 and 7. For example, the dimming/brightening trends In the NW province (Xinjiang) have the same size as the trends in

the eastern coastal provinces shown in Fig 2, but the urbanization level, speed, population, and population density in Fig 7 are very different between Xinjiang and eastern coastal provinces.

Reply 21: Thanks for the comment. In Lines 16-17, we stated that "Comparing Fig. 2A and 2B with Fig. 7, exceptions of increasing sunshine duration trends in the dimming phase mainly distribute in less-urbanized provinces, while strong declines in sunshine duration generally occur in the urbanized provinces."  Seen from Fig. 2A and 2B, increasing trends mainly occurred in southwest and northeast China, which is relatively less urbanized (Fig. 7); while the strongest decreasing trends mainly occurred in southeast China, which is relatively more urbanized (Fig. 7). Generally, that's a nationwide dimming trend. This is why both Xinjiang and the eastern coastal provinces are dominated by decreasing sunshine duration trends. But it can still be noted from Fig.2 that there are more exceptions of increasing sunshine duration trends in the Xinjiang Province, while the magnitude of declines in sunshine duration in the eastern coastal provinces is greater. The relationship between urbanization and the difference in urban and rural dimming was further illustrated in Fig. 8. To avoid misunderstandings, Lines 16-17 will be modified into:

"Comparing Fig. 2A and 2B with Fig. 6, exceptions with increasing sunshine duration trends in the dimming phase mainly distribute in less-urbanized regions (southwest and northeast parts), while strong declines in sunshine duration generally occur in the urbanized regions (southeast part)."

22. Page 8, line 14-15, (1): no air pollution emissions are shown in the paper.

Reply 22: Table 1 shows that the urbanization process is accompanied by the increase in energy consumption, industrial GDP (Gross Domestic Product) and civil vehicles, which are the major anthropogenic sources of air pollution emissions.

23. Page 8, line 15, (2): is this a conclusion?

Reply 23: The two aspects were given to correctly understand the urbanization impact, as in most cases only the second aspect was considered.

24. Page 8, line 19: clarify what "a large overlap" is.

Reply 24: According to your comments, this sentence will be modified into:

"The rates of sunshine duration changes between the selected rural and urban stations across China largely overlap by 86%."

25. Page 8, line 10-21: How big is the region to see the "regional phenomenon"? Certainly it is not the entire China.

Reply 25: Here "regional" means occurring in both urban and rural stations in the same regions. Please refer to the reply to your major comment #6.

26. Page 8, line 28-29: "As a consequence..." Not true. The apparent insignificant trend is the consequence of averaging half positives and half negative trends together to make the overall mean trend flat. There is no solid analysis provided in this paper showing the effectiveness of the air pollution control on the sunshine duration.

Reply 26: We agree with you that there is still room to further discuss the urbanization effect, for example by analyzing the trends in the different regions in China and further exploring the effectiveness of the laws as the reviewer suggested. Discussion on this will be added at the end of the fourth paragraph of Section 3.2. But both of them cannot deny the significance of this study as a first try to understanding and quantifying urbanization effect also since 1990. The overall leveling off trend after 1990 seems to stem from an offset by decreasing and increasing trends, but both declines and increases are nationwide located. Unfortunately, information on the effectiveness of the laws are lacking, but providing the number of relevant laws is already a fresh new information, which cannot be found in other studies of urbanization effects on the global dimming and brightening phenomenon.

Reference

Alpert, P., Kishcha, P., Kaufman, Y. J., and Schwarzbard, R.: Global dimming or local dimming?: Effect of urbanization on sunlight availability, Geophys. Res. Lett., 32, L1780210.1029/2005gl023320, 2005.
Alpert, P., and Kishcha, P.: Quantification of the effect of urbanization on solar dimming, Geophys. Res. Lett., 35, 2008.
Hausfather, Z., Menne, M. J., Williams, C. N., Masters, T., Broberg, R., and Jones, D.: Quantifying the effect of urbanization on U.S. Historical Climatology Network temperature records, Journal of Geophysical Research Atmospheres, 118, 481-494, 2013.
Jaswal, A. K.: Sunshine duration climatology and trends in association with other climatic factors over India for 1970–2006, Mausam, 60, 437-454, 2009.
Liley, J. B.: New Zealand dimming and brightening, J. Geophys. Res., 114, D00D10, 10.1029/2008jd011401, 2009.
Manara, V., Brunetti, M., Celozzi, A., Maugeri, M., Sanchez-Lorenzo, A., and Wild, M.: Detection of dimming/brightening in Italy from homogenized all-sky and clear-sky surface solar radiation records and underlying causes (1959–2013), Atmos. Chem. Phys., 16, 11145-11161, 10.5194/acp-16-11145-2016, 2016.
Qian, Y., Kaiser, D. P., Leung, L. R., and Xu, M.: More frequent cloud-free sky and less surface solar radiation in China from 1955 to 2000, Geophys. Res. Lett., 33, L0181210.1029/2005gl024586, 2006.
Qian, Y., Wang, W., Leung, L. R., and Kaiser, D. P.: Variability of solar radiation under cloud-free skies in China: The role of aerosols, Geophys. Res. Lett., 34, L1280410.1029/2006gl028800, 2007.
Sanchez-Lorenzo, A., Brunetti, M., Calbo, J., and Martin-Vide, J.: Recent spatial and temporal variability and trends of sunshine duration over the Iberian Peninsula from a homogenized data set, J. Geophys. Res.-Atmos., 112, D20115, D2011510.1029/2007jd008677, 2007.

Sanchez-Lorenzo, A., Calbo, J., and Martin-Vide, J.: Spatial and temporal trends in sunshine duration over Western Europe (1938–2004), J. Clim., 21, 6089-6098, 10.1175/2008jcli2442.1, 2008.

Sanchez-Romero, A., Sanchez-Lorenzo, A., Calbó, J., González, J. A., and Azorin-Molina, C.: The signal of aerosol-induced changes in sunshine duration records: A review of the evidence, Journal of Geophysical Research: Atmospheres, 119, 2013JD021393, 10.1002/2013JD021393, 2014.

Shi, G., Hayasaka, T., Ohmura, A., Chen, Z., Wang, B., Zhao, J., Che, H., and Xu, L.: Data quality assessment and the long-term trend of ground solar radiation in China, Anglais, 47, 1006-1016, 2008.

Stanhill, G., and Cohen, S.: Solar radiation changes in the United States during the Twentieth Century: Evidence from sunshine measurements, J. Clim., 18, 1503-1512, 2005.

Stanhill, G., and Cohen, S.: Solar radiation changes in Japan during the 20th century: Evidence from sunshine duration measurements, J. Meteorol. Soc. Jpn., 86, 57-67, 2008.

Tang, W. J., Yang, K., Qin, J., Cheng, C. C. K., and He, J.: Solar radiation trend across China in recent decades: a revisit with quality-controlled data, Atmospheric Chemistry and Physics, 11, 393-406, 10.5194/acp-11-393-2011, 2011.

Wang, K., Dickinson, R., Wild, M., and Liang, S.: Atmospheric impacts on climatic variability of surface incident solar radiation, Atmos. Chem. Phys, 12, 9581-9592, 2012a.

Wang, K., Ma, Q., Wang, X., and Wild, M.: Urban impacts on mean and trend of surface incident solar radiation, Geophys. Res. Lett., 41, 4664-4668, 2014a.

Wang, Y., Yang, Y., Zhou, X., Zhao, N., and Zhang, J.: Air pollution is pushing wind speed into a regulator of surface solar irradiance in China, Environmental Research Letters, 9, 054004, 2014b.

Wang, Y., and Wild, M.: A new look at solar dimming and brightening in China, Geophys. Res. Lett., 2016.

Wang, Y. W., Yang, Y. H., Zhao, N., Liu, C., and Wang, Q. X.: The magnitude of the effect of air pollution on sunshine hours in China, J. Geophys. Res.-Atmos., 117, D00V14, D00v1410.1029/2011jd016753, 2012b.

Wang, Y. W., and Yang, Y. H.: China's dimming and brightening: evidence, causes and hydrological implications, Ann. Geophys., 32, 41-55, 10.5194/angeo-32-41-2014, 2014.

Wild, M., Gilgen, H., Roesch, A., Ohmura, A., Long, C. N., Dutton, E. G., Forgan, B., Kallis, A., Russak, V., and Tsvetkov, A.: From dimming to brightening: Decadal changes in solar radiation at Earth's surface, Science, 308, 847-850, 10.1126/science.1103215, 2005.

Xia, X. A.: Spatiotemporal changes in sunshine duration and cloud amount as well as their relationship in China during 1954–2005, J. Geophys. Res.-Atmos., 115, D00k0610.1029/2009jd012879, 2010.